# Influence of weather situation on non-CO$_2$ aviation climate effects: The REACT4C Climate Change Functions

Christine Frömming[1], Volker Grewe[1,2], Sabine Brinkop[1], Patrick Jöckel[1], Amund S. Haslerud[3], Simon Rosanka[1,2,4], Jesper van Manen[1,2,5], and Sigrun Matthes[1]

[1]Deutsches Zentrum für Luft- und Raumfahrt, Institut für Physik der Atmosphäre, Oberpfaffenhofen, Germany
[2]Delft University of Technology, Aerospace Engineering, Section Aircraft Noise and Climate Effects, Delft, Netherlands
[3]Center for International Climate and Environmental Research - Oslo (CICERO), Oslo, Norway
[4]now at Forschungszentrum Jülich GmbH, Institute of Energy and Climate Research, IEK-8: Troposphere, Jülich, Germany
[5]now at Ministry of Infrastructure and Water Management, The Hague, Netherlands

**Correspondence:** C. Frömming (christine.froemming@dlr.de)

**Abstract.** Emissions of aviation include CO$_2$, H$_2$O, NO$_x$, sulfur oxides and soot. Many studies have investigated the annual mean climate impact of aviation emissions. While CO$_2$ has a long atmospheric residence time and is almost uniformly distributed in the atmosphere, non-CO$_2$ gases, particles and their products have short atmospheric residence times and are heterogeneously distributed. The climate impact of non-CO$_2$ aviation emissions is known to vary with different meteorological background situations. The aim of this study is to systematically investigate the influence of characteristic weather situations on aviation climate effects over the North Atlantic region, to identify the most sensitive areas and potentially detect systematic weather related similarities. If aircraft were re-routed to avoid climate-sensitive regions, the overall aviation climate impact might be reduced. Hence, the sensitivity of the atmosphere to local emissions provides a basis for the assessment of weather related, climate optimized flight trajectory planning. To determine the climate change contribution of an individual emission as function of location, time and weather situation, the radiative impact of local emissions of NO$_x$ and H$_2$O to changes in O$_3$, CH$_4$, H$_2$O and contrail-cirrus was computed by means of the ECHAM5/MESSy Atmospheric Chemistry model. 4-dimensional climate change functions (CCFs) were derived thereof. Typical weather situations in the North Atlantic region were considered for winter and summer. Weather related differences in O$_3$-, CH$_4$-, H$_2$O-, and contrail-cirrus-CCFs were investigated. The following characteristics were identified: Enhanced climate impact of contrail-cirrus was detected for emissions in areas with large scale lifting, whereas low climate impact of contrail-cirrus was found in the area of the jet stream. Northwards of 60°N contrails usually cause climate warming in winter, independent of the weather situation. NO$_x$ emissions cause a high positive climate impact if released in the area of the jet stream or in high pressure ridges, which induces a south- and downward transport of the emitted species. Whereas NO$_x$ emissions at, or transported towards high latitudes, cause low or even negative climate impact. Independent of the weather situation, total NO$_x$ effects show a minimum at ∼250 hPa, increasing towards higher and lower altitudes, with generally higher positive impact in summer than in winter. H$_2$O emissions induce a high climate impact when released in regions with lower tropopause height, whereas low climate impact occurs for emissions in areas with higher tropopause height. H$_2$O-CCFs generally increase with height, and are larger in winter than in summer. The CCFs of all individual species can be combined, facilitating the assessment of total climate impact of aircraft trajectories considering CO$_2$ and

spatially and temporally varying non-CO$_2$ effetcs. Furthermore they allow the optimization of aircraft trajectories with reduced overall climate impact. This also facilitates a fair evaluation of tradeoffs between individual species. In most regions NO$_x$ and contrail-cirrus dominate the sensitivity to local aviation emissions. The findings of this study recommend, to consider weather related differences for flight trajectory optimization in favour of reducing total climate impact.

# 1   Introduction

Emissions of aviation include carbon dioxide (CO$_2$), water vapour (H$_2$O), nitrogen oxides (NO$_x$), sulfur oxides (SO$_x$), and soot. Furthermore, aviation emissions cause the formation of contrails and contrail-cirrus. CO$_2$ and H$_2$O are greenhouse gases, its emissions cause a climate warming. Emissions of NO$_x$ cause a short-term production of ozone (O$_3$) and a long-term reduction of methane (CH$_4$). Changes of the O$_3$ precursor CH$_4$ cause a secondary, long-term reduction of O$_3$, which is called primary mode ozone effect (PMO). Both, O$_3$ and CH$_4$, are greenhouse gases, for which an enhancement causes a climate warming, while a reduction causes a cooling. Contrails form when hot and moist exhaust mixes with ambient air, while they only persist if the ambient air is saturated with respect to ice. Persistent contrails may evolve into contrail-cirrus which have a lifetime of up to many hours. On average, contrails and contrail-cirrus cause a climate warming, however in certain situations, they can also cause a cooling. Emissions of aerosols or aerosol precursors have a direct effect on climate, which can be warming (soot) or cooling (SO$_x$). Aviation aerosol emissions also have an indirect effect on clouds, which is still uncertain, thus the effect of aerosols has not been included in the present study. The most prominent climate effects of aviation emissions, their climate impact in terms of radiative forcing, its uncertainty, and the level of scientific understanding have been summarized by e.g. Lee et al. (2010), Brasseur et al. (2016), Lee et al. (2021).

The CO$_2$ emissions from aviation contribute 1.6% to the net anthropogenic effective radiative forcing. Taking the non-CO$_2$ effects into account, aviation constitutes 3.5% of the total anthropogenic climate impact (in terms of effective radiative forcing, Lee et al. (2021)). Compared with other modes of transportation, 12% of the CO$_2$ emissions from global transportation are caused by aviation (Brasseur, 2008). This share is expected to increase, as the aviation sector is growing at an annual rate of 1.1% over the last decade (Lee et al., 2021), while other sectors reduce their CO$_2$-emissions. However, future emissions are uncertain and the uncertainty increased due to reduced operations because of the COVID-19 pandemic (e.g. Friedlingstein et al., 2020; Le Quere et al., 2020). The Paris Agreement set the ambitious goal to keep the global temperature rise within this century well below 2°Celsius compared to pre-industrial levels. In regard of this ambitious goal, political and societal pressure for sustainable aviation increases. Mitigation options, which may enable the aviation sector to reduce its climate impact ought to be identified and evaluated.

Mitigation options may involve technological measures, such as alternative fuels, novel engine concepts, modification of aircraft design, or policy measures such as emission trading or emission reduction schemes, for instance the EU ETS (Emissions Trading System, EuropeanCommision (2015)) or CORSIA (Carbon Offsetting and Reduction Scheme for International Aviation, ICAO (2020)), or technology targets like the ACARE (Advisory Council for Aviation Research and Innovation in Europe)

Vision 2020 and FlightPath2050 (ACARE, 2020). Another efficient possibility for mitigation may be operational measures e.g. identifying alternative flight trajectories with reduced climate impact. $CO_2$ with its long atmospheric residence time is almost uniformly distributed in the atmosphere, and its climate impact is independent of the location and situation during its release.

The non-$CO_2$ gases, particles and their products, which comprise 2/3 of the aviation net effective radiative foring (Lee et al., 2021), however, have shorter atmospheric residence times and are heterogeneously distributed in the atmosphere. More precisely, the effects of non-$CO_2$ emissions depend on chemical and meteorological background conditions during their release, which vary with geographic location (e.g. Köhler et al., 2013), altitude (e.g. Frömming et al., 2012), time, local insolation (e.g. Gauss et al., 2006), actual weather, etc. Thus, regions and times which are more sensitive to non-$CO_2$ aviation emissions can

be identified. If aircraft trajectories avoid these areas with enhanced sensitivity, aviation climate impact can potentially be mitigated (e.g. Matthes et al., 2012; Grewe et al., 2014b). Such operational measures might be implemented much faster than technological improvements, which require much more time for research, development and implementation.

Previous studies investigated the annual or seasonal mean impact on contrail formation and related radiative forcing by permanent changes in flight altitudes or lateral changes of flight routes (e.g. Sausen et al., 1998; Fichter et al., 2005; Rädel and Shine,

2008; Frömming et al., 2012; Matthes et al., 2021). Others tried to avoid contrails and contrail cirrus by situation-related small changes in flight levels when flying through contrail regions (e.g. Mannstein et al., 2005; Chen et al., 2012), or they calculated horizontal flight trajectory changes to reduce travel time through contrail formation regions (Sridhar et al., 2011). These studies found considerable potential for the reduction of contrails but related tradeoffs were considered only by means of different metrics (e.g. fuel consumtion versus travel time through contrail regions) or not at all. Irvine et al. (2014) presented a framework

for the consistent assessment of maximum extra distance to be added to a flight for avoiding contrails without generating an increase in overall climate impact, finding a high dependency on the metric, time horizon and aircraft type. Zou et al. (2016) considered both, horizontal and vertical aircraft trajectory changes, and minimized the total flying cost of fuel consumption, $CO_2$ emissions, travel time and contrail formation by converting the climate impacts and other resulting tradeoffs into monetary value. Climate impacts were considered in terms of Global Warming Potential (GWP), thus a strong dependence on the time

horizon was found. A study by Hartjes et al. (2016) determined 3-dimensional aircraft trajectories while minimizing contrail formation and found vertical trajectory adjustements to be preferable over horizontal trajectory changes. By now, the spatial and temporal variability of climate effects from aircraft emissions of $NO_x$ on $O_3$ and $CH_4$ has mainly been considered in terms of climatological effects by means of permanent changes of flight altitudes or routes (Gauss et al., 2006; Köhler et al., 2008; Fichter, 2009; Frömming et al., 2012; Köhler et al., 2013; Matthes et al., 2021). Grewe and Stenke (2008) and Fichter (2009)

systematically investigated annual mean effects of unified emissions on $O_3$, $CH_4$, $H_2O$ and contrails in terms of their altitudinal and latitudinal dependency, by identifying regions where emissions have the largest impact in a climatological sense.

However, none of these previous studies, considered the impact of various aviation effects in relation to the actual weather situation, location and altitude in detail. Within the project REACT4C (https://www.react4c.eu/) the feasibility of optimizing flight altitudes and flight routes for minimum climate impact was explored in dependency of the actual weather situation

(Matthes et al., 2012). The North Atlantic region was chosen as a study domain because flight trajectories are not as constrained as over the continents. There are long distance flights which allow studying detours and sufficient air traffic making it worth-

while to study re-routing. Furthermore, the traffic is not too dense to enable re-routing without generating too many conflicts with other flights. Additionally this study region is characterised by synoptical scale archetypical weather patterns, which allow creating a set of representative weather situations. Initial studies of the REACT4C project established the consideration of different weather situations with respect to aviation climate impact. Irvine et al. (2013) defined characteristic weather patterns in the North Atlantic and defined proxies for the climate impact of carbon dioxide, ozone, water vapour and contrails to facilitate climate optimal aircraft routing. Grewe et al. (2014a) presented and verified the modelling concept of REACT4C, introducing the calculation of climate change functions (CCFs), which are a measure for the climate impact of a local emission (see Section 2). Grewe et al. (2014b) displayed local aviation climate effects exemplarily, pointed out consequences for climate optimized aircraft trajectories and showed the climate impact reduction potential for one specific weather pattern. Rosanka et al. (2020) presented a process-based analysis of weather-dependent aviation $NO_x$- effects. The present study covers the entire ensemble of REACT4C climate change functions (CCFs) for 8 representative weather situations in the North Atlantic region for winter and summer. The altitude, location and weather dependency of aviation climate effects of $O_3$, $CH_4$, $H_2O$ and contrail-cirrus are presented, discussed and validated. The distribution of high or low sensitivity to aviation emissions for all species is systematically examined. Thus, the present study respresents the fundamental data basis of most REACT4C-related studies. Further associated studies are based on the ensemble of these CCFs, e.g. for the development of more generic algorithmic CCFs (see Section 6), which enable the calculation of climate impact by means of meteorological key parameters (e.g. Matthes et al., 2017; Yin et al., 2018b; Van Manen and Grewe, 2019). Subsequent studies complete the range of REACT4C-related publications focussing on the climate impact reduction potential utilising either REACT4C-CCFs or algorithmic CCFs for operational mitigation through climate optimized aircraft trajectories (e.g. Grewe et al., 2017; Matthes et al., 2020; Lührs et al., 2021).

In the present study, first the methodology of calculating the weather related impact of a local emission on climate in a comprehensive climate chemistry model is presented (Section 2). The weather situations which were used in the present study are described in Section 3. An overview on the experimental set up is given in Section 2.5. The resulting climate change functions (CCFs) are presented in Section 4. The results are discussed and an outlook is given how the CCFs could be used for planning of climate optimized aircraft trajectories (Section 5). Section 6 concludes with a short summary and ideas for future studies.

## 2   Model description

The ECHAM/MESSy Atmospheric Chemistry Model (EMAC, Jöckel et al. (2010, 2016)) is a numerical chemistry climate model system which implements submodels describing physical and chemical atmospheric processes, ranging from the troposphere up to the middle atmosphere and their interactions with the biosphere, hydrosphere and geosphere. The Modular Earth Submodel System (MESSy) couples the various submodels to the core atmospheric model ECHAM5 (Roeckner et al., 2006). The chemistry is calculated by the submodel MECCA (v3.2, Sander et al. (2011)). Non-methane hydrocarbon (NMHC) chemistry is employed, reproducing the main features of the tropospheric chemistry (Houweling et al., 1998). A Lagrangian transport scheme is employed within this study using the submodel ATTILA (Reithmeier and Sausen, 2002; Brinkop and Jöckel,

**Table 1.** Overview of the time region grid points, where standardised emissions are released and specific local climate change functions are calculated.

| latitudes [°N] | 80 | 60 | 50 | 45 | 40 | 35 | 30 |
|---|---|---|---|---|---|---|---|
| longitudes [°W] | 75 | 60 | 45 | 30 | 15 | 0 | |
| pressure levels [hPa] | 200 | 250 | 300 | 400 | | | |

2019). Further, two newly developed submodels were employed within the present study, the submodels AIRTRAC and CON-TRAIL (Frömming et al., 2014). EMAC is used in a T42L41 spectral resolution, corresponding to a quadratic Gaussian grid of $\sim 2.8° \times 2.8°$ in latitude and longitude, and 41 vertical layers from the surface to 5 hPa, which is a compromise between the level of detail within the simulation and the computational expense.

## 2.1   Model setup for calculating climate change functions

Within this study, EMAC is employed to calculate the atmospheric impact of standardised air traffic emissions to the chemical composition of the atmosphere and to the formation of contrails and contrail-cirrus at predefined longitudes, latitudes, altitudes and times. The location- and time-dependent specific climate impact per emission is referred to as climate change functions (CCFs). The CCFs are a measure for the sensitivity of a certain emission location to aviation climate impact as introduced by Matthes et al. (2012). These CCFs enable the assessment of re-routing options with reduced climate impact. Hence, in this study we present a numerical modelling approach for calculating climate change functions with the modular global chemistry climate model EMAC. In order to quantify this spatially and temporally dependent climate impact information, it is necessary to calculate the atmospheric impact in terms of concentration changes and associated radiative impact of a specific emission at the given geographic position, altitude and time of emission. We use a Lagrangian scheme (ATTILA) in the modular Earth system model EMAC in order to numerically simulate the fate of emissions on Lagrangian trajectories and quantify associated atmospheric impacts, concentration changes and radiative impacts in terms of radiative forcing. This radiative forcing is translated to a climate metric in order to quantify the climate impact. By remapping these impacts quantified with EMAC to the specific location and time of emission, we construct 4-dimensional climate change functions. For a detailed description of the methodology we refer to the companion model development paper of Grewe et al. (2014a), while in the present study we resume only the relevant information and focus on the results.

## 2.2   Chemical changes

To derive the CCFs, a 4-dimensional time-region grid is defined. This time-region grid covers cruise altitude relevant pressure levels from 400 hPa to 200 hPa over the North-Atlantic area (between 30 and 80°N and between 80 and 0°W), in total yielding 168 grid points (see Table 1). At each of these time-region grid points, a pulse emission of $NO_x$ ($5 \times 10^5$ kg NO) and $H_2O$

($1.25 \times 10^7$ kg $H_2O$) is released by means of the submodel TREXP (Tracer Release EXperiments from Point sources) within one model timestep of 15 minutes. A Lagrangian aproach was chosen as it facilitates the calculation of CCFs for a number of time-regions within a single EMAC simulation. The tracer concentration from the pulse emission is equally distributed on 50 air parcel trajectories, which are (randomly distributed) started from within the EMAC grid box in which the time-region grid point lies. The air parcel trajectories are advected by the submodel ATTILA using the wind field from EMAC. Diffusive processes through inter-parcel mixing of air parcel trajectories with tracer loading with e.g. empty background trajectories ($\sim$169 000 in the Northern Hemisphere) are parameterized by means of the submodel LGTMIX. The mixing of adjacent parcels is represented by slightly changing the mass mixing ratio $c_i$ in a parcel towards the average mixing ratio of all parcels within one grid box $\bar{c}$. The new mixing ratio of the air parcel is then calculated by $c_i^{new} = c_i + (\bar{c} - c_i)d$, with $d$ being a dimensionless mixing parameter within the range [0,1], controlling the magnitude of exchange (Reithmeier and Sausen, 2002; Brinkop and Jöckel, 2019). $NO_x$ emissions influence the production and loss of $O_3$, and the loss of $CH_4$ via the partitioning of OH and $HO_2$. For each air parcel trajectory, the contribution of the pulse emission to the atmospheric composition is calculated by the newly developed submodel AIRTRAC (Frömming et al., 2014). AIRTRAC solves a simplified set of chemical equations on the air parcel trajectories (see Grewe (2013)) by using the diagnosed production (P) and loss (L) terms from the kinetic solver (MECCA) of the background chemistry, while the proportional contributions of the emitted species to the atmospheric mixing ratios of $NO_y$ (all active nitrogen species), $HNO_3$, $O_3$, $HO_2$, OH, and $CH_4$ are calculated for each air parcel trajectory. Therefore the emissions need to be partitioned into background emissions (b) and additional emissions (e). The ozone production for additional emissions ($P_{O_3}^e$) via reaction

$$HO_2 + NO \rightarrow OH + NO_2 \tag{R1}$$

is calculated (according to Grewe et al. (2010)):

$$P_{O_3}^e = P_{O_3}^b \cdot \frac{1}{2}\left(\frac{HO_2^e}{HO_2^b} + \frac{NO^e}{NO^b}\right) \tag{1}$$

The reaction rate for ozone loss from additional emissions ($L_{O_3}^e$) via reaction

$$NO_2 + O_3 \rightarrow NO + 2O_2 \tag{R2}$$

is calculated similarly:

$$L_{O_3}^e = L_{O_3}^b \cdot \frac{1}{2}\left(\frac{NO_2^e}{NO_2^b} + \frac{O_3^e}{O_3^b}\right). \tag{2}$$

This exemplarily demonstrates the methodology of calculating the atmospheric changes attributed to the aditionally emitted species, further details are elaborated in Grewe et al. (2014a). An analogous approach is used for $CH_4$, while taking into account the most relevant reactions with regard to OH and $HO_2$. Atmospheric processes such as wash-out and dry deposition, are also proportionally taken into account on the air parcel trajectories. For $H_2O$ emissions only loss processes are considered. The loss processes affecting the additionally emitted $H_2O$ are included proportionally to the precipitation rate (Grewe et al., 2014b). Figure 1 shows exemplarily the temporal evolution of the contributions from NOx emissions for one arbitrary time-region to the

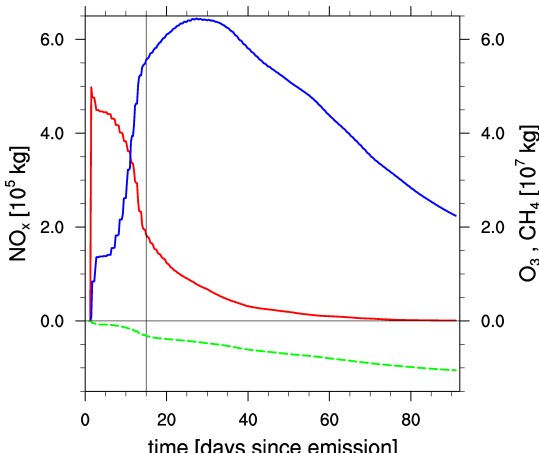

**Figure 1.** Temporal evolution of the contribution to the global atmospheric mass of $NO_x$ (red), $O_3$ (blue) and $CH_4$ (green) attributed to the emissions in one arbitrary time-region within 90 days after emission.

$NO_x$, $O_3$ and $CH_4$ atmospheric burden. $NO_x$ is completely washed out within the first 30-40 days. $O_3$ increases while additional $NO_x$ is available, then the $O_3$ loss dominates and $O_3$ reduces gradually. At the same time $CH_4$ is reduced due to an increase in OH, which is caused by additional $NO_x$ first and by additional $O_3$ later. The calculation of chemical contributions from time-region emissions are accomplished on the air parcel trajectories, i.e. in Lagrangian space. The background information required for the chemical calculation are transferred from grid point space to Lagrangian space.

### 2.3 Contrails and contrail-cirrus

Contrails may form in the atmosphere, if the ambient air is cold and moist enough, and they may persist, if the air is supersaturated with respect to ice (Schumann, 1996) and evolve into contrail-cirrus. Within this study we do not distinguish between linear contrails and contrail cirrus. In EMAC the atmospheric ability to form persistent contrails and contrail-cirrus is calculated at each timestep according to Burkhardt et al. (2008) and Burkhardt and Kaercher (2009). This atmospheric ability is referred to with the term potential contrail coverage. The potential contrail coverage indicates the fraction of a grid box which can be covered by contrails at maximum. The potential contrail coverage is transferred onto the air parcel trajectories, so are the background properties required for the contrail calculations. Then the actual contrail coverage is determined in dependency whether air traffic occurs in the respective grid box. This is true for the respective time-region, where one pulse of air traffic emissions initiates the formation of contrails. Their temporal development according to spreading, sublimation and sedimentation of ice particles is parameterized as follows. The prognostic equation for contrail coverage is the sum of newly formed contrails and spreaded contrails from preceding timesteps:

$$\frac{db}{dt} = \left(\frac{db}{dt}\right)_{new} + \left(\frac{db}{dt}\right)_{spread} \tag{3}$$

The coverage of newly formed contrails is determined from initial contrail dimensions (width and length of the contrail in EMAC gridbox) while the spreading is parameterised depending on the vertical wind shear according to Burkhardt and Kaercher (2009) (see Grewe et al. (2014a)). Contrails are characterized through their coverage ($b$) and their water mixing ratio ($m$). The prognostic equation for contrail ice water mixing ratio $m$ includes the formation of new contrails, sedimentation of ice, depo-

sition of water vapour on contrail ice particles, and sublimation Burkhardt and Kaercher (2009).

$$\frac{dm}{dt} = \left(\frac{dm}{dt}\right)_{new} + \left(\frac{dm}{dt}\right)_{sed} + \left(\frac{dm}{dt}\right)_{dep/subl} \tag{4}$$

The newly formed ice water depends on the condensation rate in the contrail-covered part of the grid box (Ponater et al., 2002), the sedimentation of ice is parameterised according to the divergence of the flux of ice particles according to Heymsfield and Donner (1990) and the sublimation and growth depends on the tendency of potential contrail coverage. Further details are given by

Grewe et al. (2014a) and Frömming et al. (2014).

## 2.4   Radiative forcing and climate change functions

The radiative transfer calculations are accomplished in gridpoint space, therefore the properties of air parcel trajectories are transferred from Lagrangian space to gridpoint space. The approach described in the previous sections and more detailed in

Grewe et al. (2014b) leads to a 4-dimensional distribution of mixing ratios of trace gases, coverage and optical properties of contrails, which are caused by a local pulse emission over an integration time of 90 days (in case of chemical perturbations) or 3 days (in case of contrails and contrail-cirrus). The 90 days cover most of the short-term responses resulting from $NO_x$ emissions, while longer term responses are projected by extrapolation. The impact of the perturbations is quantified by means of radiative forcing (RF), the radiation imbalance at the tropopause caused by radiatively active species (Shine et al., 1990).

Positive RF will lead to climate warming and vice versa. The instantaneous radiative forcing at the tropopause is calculated directly within the submodel RAD4ALL (Dietmüller et al., 2016) for $O_3$, $H_2O$ and contrails. The stratosphere-adjusted RF, which allows stratospheric temperatures to adjust to the new equilibrium following the radiative imbalance, is derived from the instantaneous RF as described in detail by Grewe et al. (2014a). The RF from $CH_4$ is determined from the $CH_4$ mass perturbation following the method described by Shine et al. (1990). The RF from primary mode $O_3$ (PMO) is derived by ap-

plying a constant factor of 0.29 to the $CH_4$ RF (Dahlmann, 2012). The adjusted $H_2O$-RF is calculated based on results from Grewe and Stenke (2008) employing their relationship between $H_2O$-mass and adjusted RF. Because of its long perturbation life time, emissions of $CO_2$ are assumed to be equally mixed within the atmosphere. The temporal evolution of the change in mixing ratio is calculated following Fuglestvedt et al. (2010) and Forster et al. (2007). The RF calculation for $CO_2$ includes a simple linearised conversion factor between the change in its atmospheric mass and the RF as given by Grewe et al. (2014a).

Regarding contrails, the difference between the adjusted and the instantaneous RF is marginal (Marquart et al., 2003), hence within the present study, the instantanous RF is used. Because of the overall setup of the experiment, in some cases very small ice water contents were simulated. Several other radiation parameterisations are limited to ice water contents or optical depths ($\tau$) exceeding a certain threshold (personal communication R. R. de Leon, MMU). Although for the model used here, no such

validity range exists, in the present study, only contrails with $\tau \geq 0.01$ are included, as it is suspected, that in some cases, very small optical depths in combination with small coverages may not yield correct short wave radiative forcings.

From RF various climate metrics can be derived by means of simple climate response functions. Different climate metrics aim to answer different political questions. As the main components of climate metrics (RF and $\Delta$ T, global mean temperature change) are accumulative and annually variable, averaged or time-integrated climate metrics are favourable. They evaluate mean impacts over an integration period and are less sensitive to lifetime. A suitable climate indicator is the averaged global mean temperature change integrated over a certain time horizon H, i.e. the Average Temperature Response (ATR, Schwartz Dallara et al. (2011)) or similarly the average integrated Absolute Global Temperature Potential (iAGTP divided by the time horizon H, Aamaas et al. (2013)). Here, we calculate the average global mean temperature change following Aamaas et al. (2013) for a pulse emission integrated over a time horizon of 20 years, as we focus on the short-term effect of a climate-optimized re-routing strategy. Based on the RF calculations, other climate metrics could be calculated for other time horizons e.g. 20, 50, or 100 years (Fuglestvedt et al., 2010), such as the absolute global warming potential (AGWP), or the absolute global temperature potential, not only for pulse but also for sustained emissions or future emission scenarios, which would give a wide range of CCFs. A temperature based climate metric has the advantage that it is both, used within the climate modeling community, but also understood by nonexperts. A discussion on the suitability of various metrics and time horizons regarding different research questions (e.g. pulse versus sustained emissions) is given by Grewe et al. (2014a) and Grewe and Dahlmann (2015).

## 2.5  Experimental setup

For the calculation of the 4-dimensional climate change functions predefined emissions are released in each time region and distributed on a set of 50 Lagrangian air parcel trajectories, respectively. The emission regions as listed in Table 1 comprise 168 locations (6 longitudes x 7 latitudes x 4 pressure levels). For each emission location the emission impact calculations were performed using the global chemistry climate model EMAC for episodic simulations. As the intention was to investigate, to what extent the local aviation climate effects change during the course of the day, three emission times (6, 12, and 18 UTC) were considered. Fifteen time-regions could be comprised within one simulation. Thus for the 4032 time-regions in total (168 grid points $\times$ 3 emission times $\times$ 8 weather situations) plus one simulation for spin-up and definition of characteristic weather patterns, a total of $\sim$270 simulations were performed, using 13 500 CPU-h. The spin-up period was 6 months, further 18 months were used for the identification of the characteristic weather situations in EMAC. The year 2000 was simulated, including assumptions for background emissions for that year (e.g. Hoor et al., 2009). The definition-period was used to match EMAC model weather situations with characteristic weather patterns as defined by Irvine et al. (2013) from ECMWF Re-Analysis Interim data. A model date was selected for each of the characteristic weather situations for winter and summer, respectively. E.g. December 23rd in 2000 in EMAC was found to match with weather pattern W1 from Irvine et al. (2013) in ERA Interim, etc. The selected dates from the definition-period were used for restarting the model to simulate the characteristic weather situation for the integration period. Model dates were chosen, where the selected weather pattern remained rather stable for the

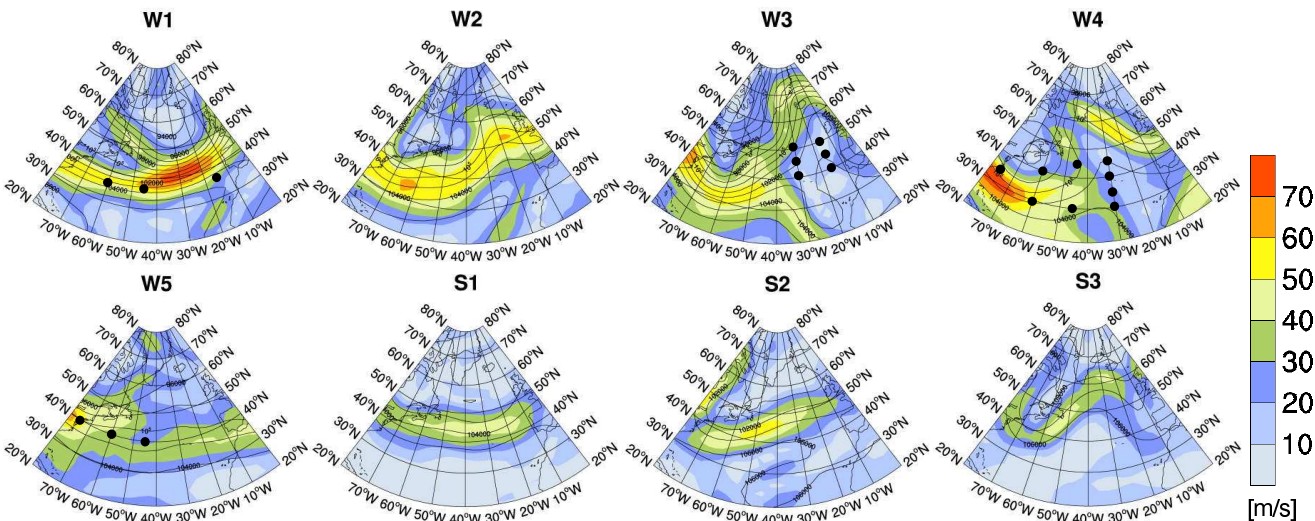

**Figure 2.** Geopotential height (black contours in gpm) and wind velocities in m/s (colourbar) at 250 hPa for five representative winter weather situations (W1–W5) and three summer situations (S1–S3) as simulated with EMAC using the classification of Irvine et al. (2013). The black dots mark the regions discussed in Section 4.2 and listed in Table A1. Please note that the maps in Figure 2 show a somewhat larger area for a better representation of the weather patterns, than the maps showing the CCFs in Figures 3, 7, 8, 11 and 12.

following 3-5 days. Then other weather situations followed. Within this study we performed quasi-chemical-transport model (QCTM) simulations Deckert et al. (2011). These simulations are dedicated to ensure that the changes from the time-region
emissions do not feed back to the base model processes and identical background meteorology and chemistry is guaranteed for the spin-up and definition phase and for all time-region simulations. The contrail RF was calculated for 3 days after the emission occured, after this period no contrails remained, while the RF for $NO_x$, $CH_4$ and $H_2O$ was calculated for 3 months after the emission occured. Most effects faded-out during that period, however, effects which were still going on were extrapolated.

## 275 3 Weather situations

Figure 2 shows eight representative weather situations in the North Atlantic as determined within the EMAC model. These typical weather situations were defined according to the classification of Irvine et al. (2013). They represent the variability in the North Atlantic in winter and summer. The patterns were determined by their similarity to the North Atlantic Oscillation (NAO) and East Atlantic (EA) teleconnection patterns and can be characterized by the strength and position of the jet stream.
Five specific types are defined in winter time. Weather situation W1 shows a strong zonal jet stream and a low pressure trough is dominating the North Atlantic. Weather situations W2 and W3 represent a meridionally tilted jet stream with either a weaker or stronger jet, respectively. Weather situation W4 is characterized by a ridge over the Eastern North Atlantic and the jet is confined to the western part of the North Atlantic. W5 shows the least similarity to the NAO and EA teleconnection patterns

and the jet is weak and confined to the US coast. W5 is the most frequent weather situation in winter (26 days per season). Types W1 to W4 occur on average 15–19 days per winter (Irvine et al., 2013). In summer only three types are defined, because of weaker teleconnection patterns and a smaller variability of the jet. Weather pattern S1 represents a strong zonal jet stream, although the jet is weaker than in winter. Weather pattern S2 is characterized by a jet, which is weakly tilted towards northeast. Weather pattern S3 shows a weak, but strongly tilted jet. Weather patterns S1 and S3 occur with similar frequency (19 and 18 days per summer, respectively). S2 is the most frequent type in summer (55 days per summer). For each of these 8 weather situations a representative model date is selected, for which the weather dependent climate change functions were calculated. Results of CCFs for one specific weather situation (W1) were exemplarily shown by Grewe et al. (2014b). Here, an overview of the climate change functions for all representative winter and summer weather situations are presented and analysed in detail, with particular focus on the differences between the weather situations.

## 4 Climate change functions

The climate change functions were calculated for all time-regions by means of episodic simulations with the global chemistry climate model EMAC. For every time-region the development of contrails, the decay of $H_2O$ and the production and loss of $O_3$, $CH_4$, PMO and other trace gases were calculated as described in Section 2 and the respective radiative forcing and the average global temperature response for a time horizon of 20 years (ATR20) were determined. This means that for every time-region and every species one global number results, which refers to the original emission location. In the figures which are shown in the following section denoted as climate change functions, these global numbers are plotted at the location of the corresponding time-region, i.e. where the original emission took place (although the colour coded value represents a global effect). In other words, if emissions are released at a time-region where CCFs show a high sensitivity (colour-coded in red in the CCF-Figures 7,8, 11, 12), these emissions cause a high global climate impact. Whereas emissions released at locations with low CCF values (e.g. light yellow or even blue in these Figures), these emissions cause low or even negative global climate impact (cooling). Within this research paper we exemplarily show CCFs for the 250 hPa level and an emission time of 12 UTC. The entire ensemble of CCFs derived within REACT4C for all species for 4 emission levels, 3 emission times and 8 weather situations are presented in the supplement.

We describe the potential contrail coverage and contrail-cirrus CCF for all weather situations in section 4.1, the CCFs for $O_3$ and the combined effects of $O_3$ and $CH_4$ (total $NO_x$) are presented in section 4.2 and the CCFs for $H_2O$ are described in section 4.3.

### 4.1 Contrail effects

Figure 3 shows the potential contrail coverage for the eight representative winter and summer weather situations as simulated with the EMAC model. The potential contrail coverage indicates the probability of atmospheric conditions enabling the

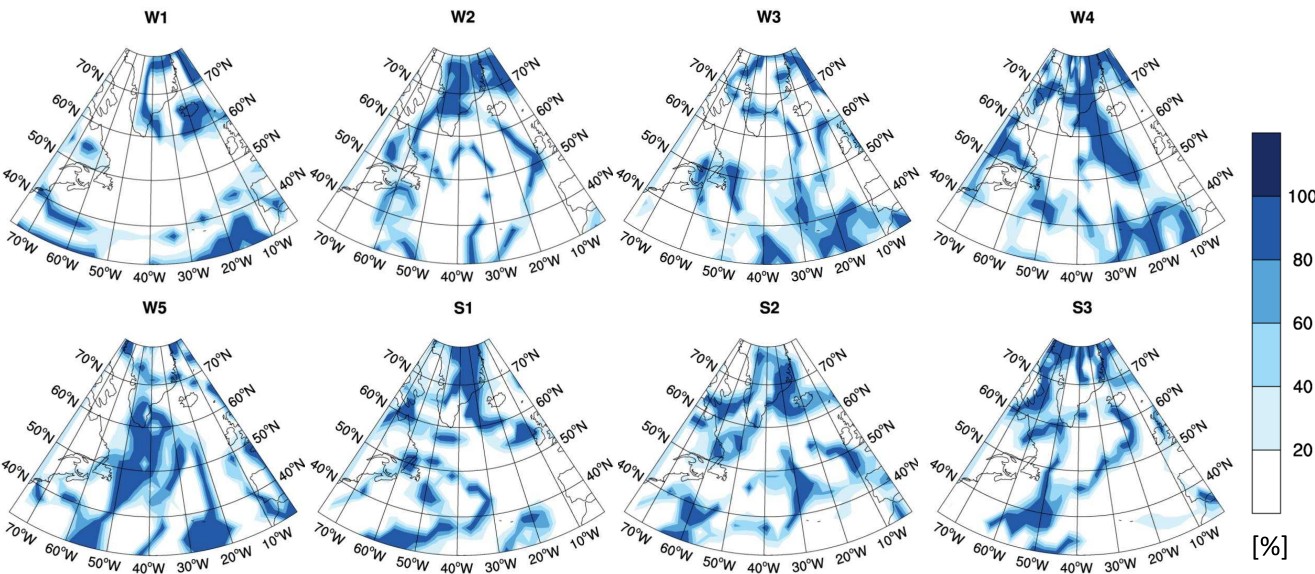

**Figure 3.** Potential contrail coverage at the timestep of emission release (12 UTC) in % at 250 hPa for eight representative weather situations.

formation of persistent contrails. A potential contrail coverage which is non-zero in the timestep of emission release in the respective time-region is a prerequisite for a non-zero contrail CCF (shown in Figure 7), only then a contrail can be formed. The magnitude and sign of the contrail CCF depends on the temporal development of contrail properties according to spreading, sublimation and sedimentation and on environmental properties e.g. natural clouds, solar insolation. When averaged over the
year (not shown) maximum potential contrail coverage is found in the stormtrack region and over Greenland, whereas minimum potential coverage is found over Newfoundland. In our study region, the mean potential contrail coverage ranges from 20% – 36% at 250 hPa between the weather situations. The actual magnitude and distribution of potential contrail coverage depends on season and weather situation. Highest potential coverages are found in weather situations W4, W5 and S2 between 250 and 300 hPa. Minimum contrail formation is found at 400 hPa independent of the weather situation, since the temperature
threshold for contrail formation is more frequently surpassed at that level, particularly at low latitudes. Common features found in the weather situations studied here, are an enhanced potential contrail coverage in the vicinity of Greenland, where saturation is induced by orographic lifting. Enhanced potential contrail coverage can be found south of the jet stream (see Figure 3 e.g. W1), where the tropopause is higher, and in areas with strong meridional transport, e.g. around ridges (e.g. W4), where air masses are lifted, which also leads to saturation. In contrast, comparably low potential coverages can be observed in the area
of the jet stream (e.g. W1, W3).

As described in section 2, aviation emissions are released in every time-region in our study area, resulting in contrail coverage if atmospheric conditions allow for it. The contrail coverage and ice water content is transported on air parcel trajectories and

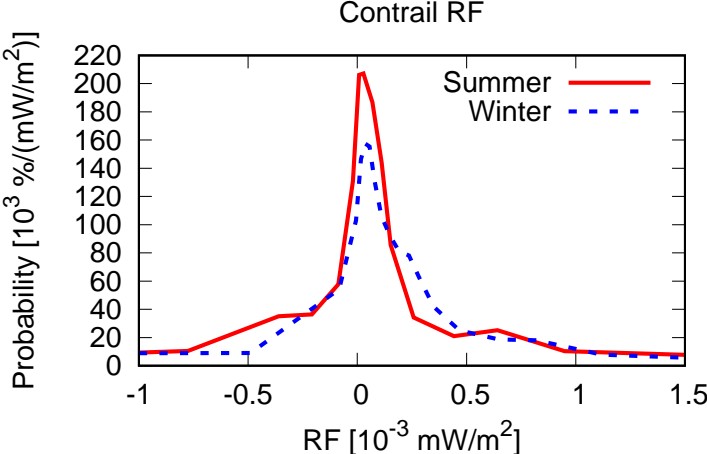

**Figure 4.** Probability density function of contrail net RF for winter and summer weather situations.

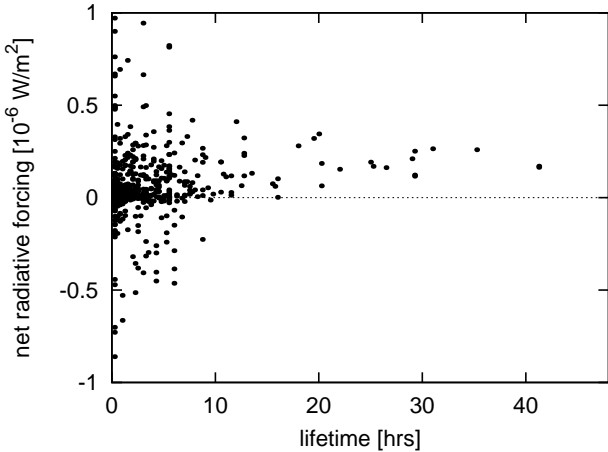

**Figure 5.** Instantaneous net radiative forcing of contrails relative to the contrail lifetime for winter and summer weather situations.

evolve according to spreading, sedimentation and sublimation. The span of contrail lifetimes ranges from 15 minutes to more than 24 h. A mean lifetime of contrails of 3.5±5.3 h was found for all weather situations considered. In winter, contrails exist on average 4.0 h, with mean lifetimes ranging from 1.9 h for W2 to over 5 h for W4 and W5. In summer, the mean lifetime of contrails is shorter (2.5 h) and similar for all weather situations. 78% of all contrails live less than 5 h, and only 7% of all contrails live longer than 10 hours. Contrail net RF is the sum of positive longwave and negative shortwave RF of similar magnitude (Ponater et al., 2002). Figure 4 shows the probability density function of contrail net RF, emphasizing that the majority of contrails within the present study causes a mean positive net RF. The scatterplot of net RF versus contrail lifetime for all contrails in Figure 5 shows that a negative net RF may only occur for contrails with lifetimes less than 10 hours independent

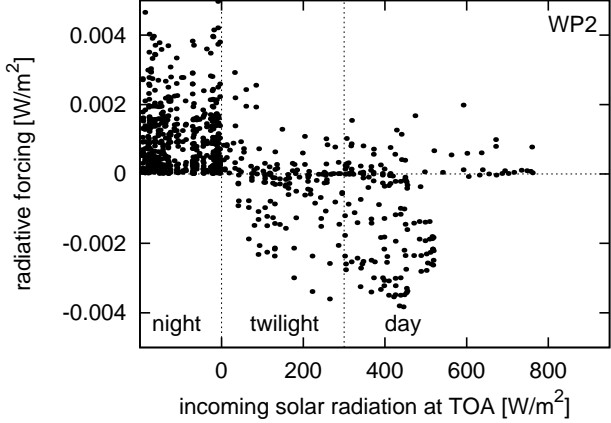

**Figure 6.** Instantaneous net radiative forcing at top of atmosphere (TOA) of individual contrails relative to the actual, local time (night, daytime, twilight) for weather pattern W2.

of the weather situation. This is due to the fact, that contrail RF may only be negative during daytime within a short timeframe (during and close to twilight, Meerkötter et al. (1999)). The strong shortwave cooling during twilight is caused by the flat an-
gle of incidence and a comparably longer path through the contrails and therefore higher reflective impact (Meerkötter et al., 1999). The longer a contrail lives, the higher is the probability, that a larger amount of its lifetime lies outside this timeframe. The scatter plot in Figure 6 shows the instantaneous net radiative forcing relative to the actual, local day-/night-time for all timesteps over the whole contrail lifetime for all time-regions for weather pattern W2. At night (net top solar radiation = 0, horizontally spread for better readability), contrails cause only positive net radiative forcing. During twilight and daytime with
low incoming solar radiation (up to $\sim$500 Wm$^{-2}$), the largest spread of net radiative forcing is found, which can be both positive or negative, resulting from positive longwave and negative shortwave RF of similar extent.

To enhance the spatial resolution of the contrail-cirrus CCFs, the contrail RF, which is initially available on the resolution of the time region grid ($15° \times 5$–$20°$), is masked with the information whether contrail-cirrus formation is possible at all (i.e.
the potential contrail coverage), which is available at the finer spatial resolution of the EMAC model ($\sim 2.8° \times 2.8°$). Figure 7 shows the climate change functions of contrail-cirrus in terms of ATR20 for all weather situations exemplarily at 250 hPa for an emission time of 12 UTC. Contrail-cirrus CCFs for other pressure levels and times are shown in the supplementary material. Overall, the CCFs of contrail-cirrus show a strong spatial and temporal variability and large differences with respect to the various weather situations. In general, the climate impact of contrail-cirrus may be both, positive or negative, and the sign of
the instantaneous radiative forcing can even change during their lifetime. On average the positive radiative forcing dominates within all weather situations, indicated by positive CCF values. In winter, all contrails northwards of 60° N have a warming climate impact as they are nighttime contrails in most cases. Similarities which indicate a significant relationship from the weather situation to the CCFs are not easily identified, although we find a few characteristic features: an enhanced climate

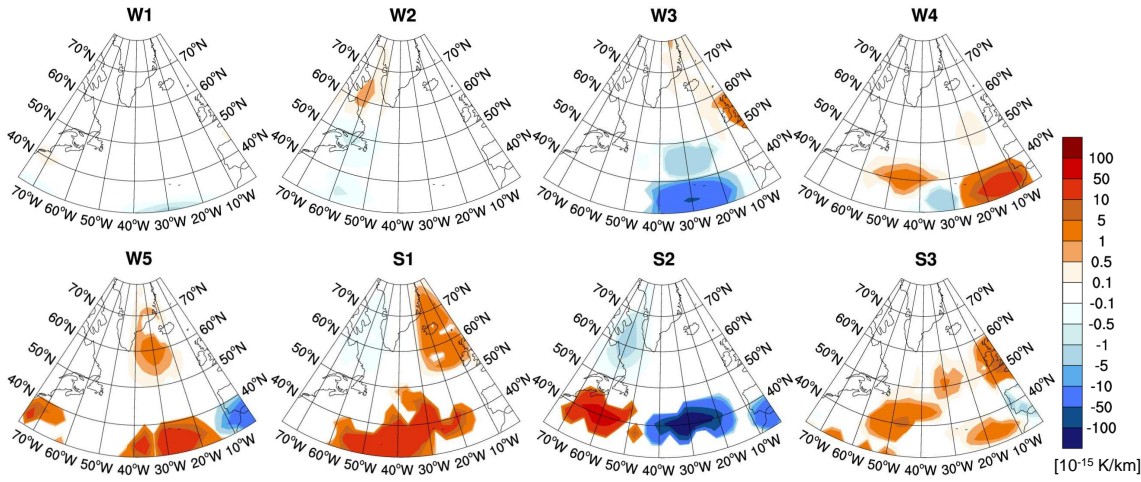

**Figure 7.** Climate change functions of contrail-cirrus at 250 hPa in $10^{-15}$ K/km(flown) for 8 representative weather situations and an emission at 12 UTC. CCFs for other pressure levels and times investigated in this study are presented in the supplement.

impact (irrespective if positive or negative) south of the jet stream, in the vicinity of Greenland and in areas with strong merid-

ional transport (e.g. around high pressure ridges in W3, W4). In contrast, close to the jet stream comparably low contrail-cirrus climate impact is found. Whether contrails have a positive or negative CCF depends mainly on the solar insolation (day/night) during their entire lifetime. If contrails have a cooling climate effect, a considerable part of their lifetime must exist during daytime. Contrails which exist only during night time only have a warming effect. Hence, also the time of emission release is essential in combination with the contrail lifetime as these parameters have the potential to change the sign of contrail climate

impact. This becomes apparent in the supplement, where contrail-cirrus CCFs for all emission times are shown. For example in W2, there is a warming contrail-cirrus area in the eastern Atlantic at 6 UTC, which is nighttime in that region. This CCF turns into a cooling contrail-cirrus area for emissions at 12 UTC (early morning hours in that region), while for emissions at 18 UTC (afternoon), the CCF becomes positive again.

**4.2   Nitrogen oxide effects**

Figure 8 shows the climate change functions due to ozone ($O_3$) changes induced by aviation $NO_x$ emissions. The $O_3$ CCF is always positive (= warming) and the spatial variabilities of the $O_3$-CCFs are significantly lower than those of contrail and contrail cirrus. The $O_3$ CCF patterns show distinct similarities to the weather pattern (Figure 2). Larger $O_3$ CCF values are found in the area of the jet stream, e.g. at 35°N, 50°W in W1, or in the area of the high pressure ridges, e.g. reaching from the

central or east Atlantic towards Iceland or Greenland in W2, W3 and W4, whereas low $O_3$ CCFs are found at high latitudes in the winter weather patterns and in the area of low pressure troughs. These similarities between the weather patterns and the

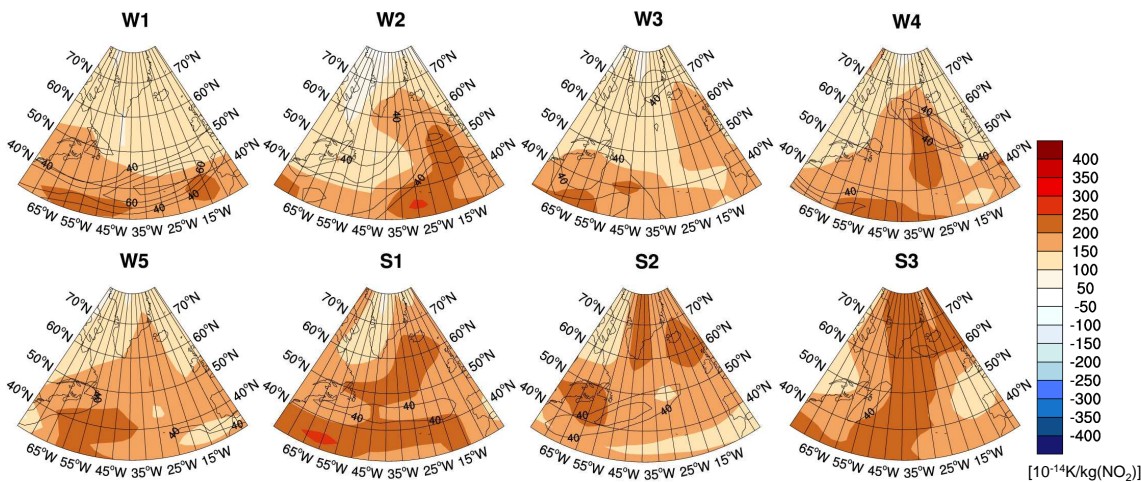

**Figure 8.** Climate change functions of aviation induced $O_3$ at 250 hPa in $10^{-14}$ K/kg($NO_2$) for 8 representative weather situations and an emission at 12 UTC. The isolines show the windspeed in m/s (>40 m/s) at 250 hPa. CCFs for other pressure levels and times investigated in this study are presented in the supplement.

CCFs patterns indicate that the meteorological situation during the time of emission strongly influences the ozone production and thus the climate impact of the emitted $NO_x$. Our findings are supported by earlier climatological studies showing a higher $O_3$ response for $NO_x$ emissions at low latitudes (e.g. Berntsen et al., 2005; Köhler et al., 2008; Grewe and Stenke, 2008).

Gauss et al. (2006) found an amplified seasonal variation of $O_3$ effects for enhanced emissions at high northern latitudes. They found a reduction of $O_3$ for enhanced use of polar routes because of reduced or even absent photochemistry at high latitudes in winter, but an increase in total $O_3$ for enhanced polar routes in summer, because an increased fraction of emissions is released in the lowermost stratosphere, where $NO_x$ accumulates more efficiently, yielding an increase in total $O_3$ burden.

To what extent different transport pathways affect the ozone production following an emission of $NO_x$ is exemplarily illustrated
in Figure 9, showing the evolution of ozone for two neighboring emission locations (A and B), both located at 45° N, but at different longitudes (30° W and 15° W, respectively) and in different relation to the high pressure ridge in weather pattern W3. The air parcel starting at emission location A (left panel of Figure 9) is west of the high pressure ridge and stays at higher altitudes (above an altitude of 350 hPa in the first weeks) and at higher latitudes (northward of 30° N). Whereas the air parcel starting at B, located in the high pressure ridge, is transported southwards and downwards after emission and stays at ~30° N
and below an altitude of ~300 hPa until the end of February. During that time the ozone mixing ratio increases strongly and remains at around 30 to 40 nmol/mol. Whereas the air parcel starting at A (west of the high pressure ridge) experiences only moderate ozone production because of the scarce availability of sunlight at high latitudes in January and February, yielding an ozone mixing ratio of only 15 to 20 nmol/mol. This example demonstrates the importance of the emission location and of the meteorological conditions during emission and subsequent transport pathways towards different chemical regimes: High
photochemical $O_3$ production occurs for transport pathways towards lower latitudes and altitudes whereas only moderate $O_3$

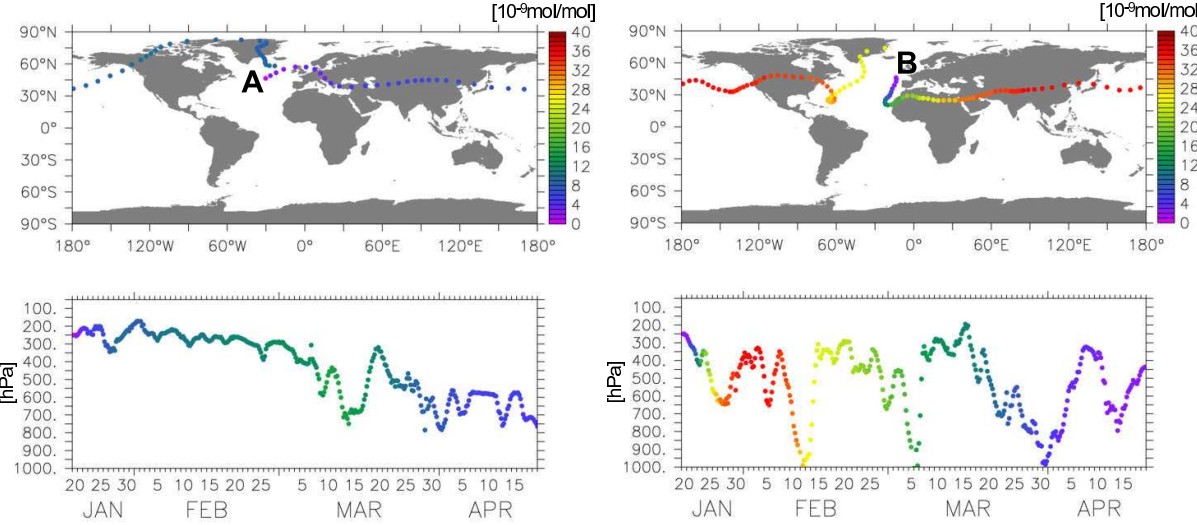

**Figure 9.** Evolution of the ozone mixing ratio along air parcel trajectories for two different emission locations in weather pattern W3: A (45° N and 30° W, left) which is west of the high pressure ridge and B (45° N and 15° W, right) which is in the high pressure ridge. Top: Geographical location of the air parcel in 6 hour intervals, the colour indicates the ozone mixing ratio [$10^{-9}$ mol/mol]. Bottom: Temporal evolution of the altitude of the air parcels. For reasons of clarity, the top panels show a shorter period than the bottom panels (A: ~15 days, B: ~27 days, i.e. the time in which the air parcel surrounded the earth once).

production is found for transport pathways towards high altitudes and latitudes.

In order to better understand how the prevailing meteorology during an emission event and the subsequent transport pathways influence the $O_3$ CCFs, we analyse in detail where and when the bulk $O_3$ increase occured for trajectories started in different meteorological situations. In the following this is referred to as the $O_3$ gain latitude, altitude and time. We identify three regions, which are exemplarily studied for W1, W3, W4 and W5, having either a pronounced high pressure ridge (W3, W4) or a zonally oriented jet stream (W1, W4, W5) in common. The regions discriminated in this analysis are:

a  In the high pressure ridge,

b  west of the high pressure ridge, and

c  at high $O_3$ CCF regions near the jet stream

(see Appendix A for their definition). Figure 10 (left) shows the probability density function of the $O_3$ gain latitude for W1, W3, W4 and W5 based on 300 and 450 trajectories for the high pressure ridge and jet stream related situations, respectively (see Tab. A1 and black dots in Figure 2 for the trajectory starting points). All three meteorological situations show a wide spread of the ozone gain latitude between 0° and 60° N. However, there is a clear difference in the pdfs of the main ozone

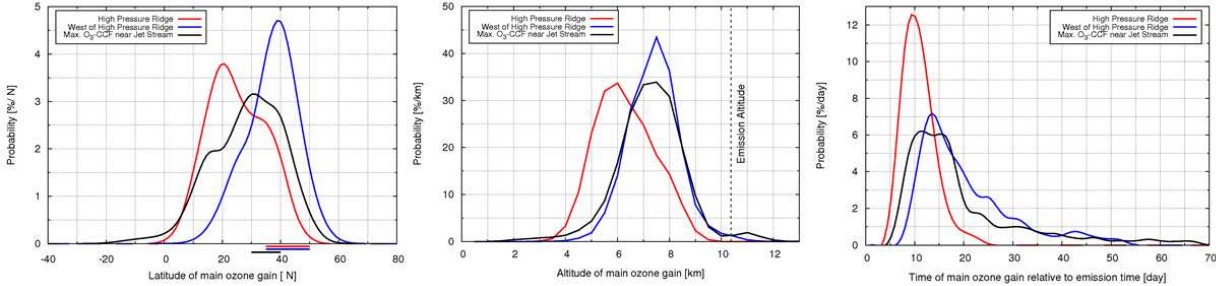

**Figure 10.** Weighted probability density function of the main ozone gain latitude (left), altitude (mid), and time (right) for 3 different regions: In the high pressure ridge (red), west of the high pressure ridge (blue), and in the area of large $O_3$-CCF values near the jet stream (black) (for information on regions and weighting see Appendix A, Tab. A1 and black dots in Fig. 2). The location of emission release is indicated by bars.

gain latitude (Figure 10 (left)) for the trajectories started in the area of the high pressure ridge with a major mode at $20°$ N (red curve) compared to the trajectories started west of the high pressure ridge with a major mode at $40°$ N (blue curve). Emissions released within the high pressure ridge have a larger probability to be transported towards the tropics compared to trajectories started at the same latitude but west of the ridge, which stay at higher latitudes or are transported even northwards. The ozone gain latitude of trajectories starting west of the high pressure ridge is close to the latitude of the emission. The ozone

gain altitude and time (Figure 10, mid and right) shows a main mode at 6 km and 10 days for trajectories starting in the high pressure ridge (red curve) and 7.5 km and ∼15 days for trajectories starting west of the ridge (blue curve), respectively. Note that the shape of the pdfs differs significantly. The air parcels starting in the high pressure ridge experience the ozone gain earlier and at lower altitudes than the air parcels starting west of the ridge, which experience ozone gain for a much longer period and at higher altitudes. As known from general meteorology, the transport pathways are controlled by the location of air

parcels relative to the Rossby waves, leading to transport into different chemical regimes, such as the tropical mid troposphere or the mid to high latitude lowermost stratosphere, which are characterised by a high or low chemical activity, respectively. In a companion paper (Rosanka et al., 2020) the interdependency of the time and magnitude of the $O_3$ maximum to the containment of air parcels within high pressure systems during emission has been further analyzed. They detected a high correlation of early $O_3$ maxima when air parcels were released within high pressure systems and found, that high $O_3$ changes were only possible

for early $O_3$ maxima. The $O_3$-CCF values in the vicinity of the jet stream are of similar magnitude as the values in the area of the high pressure ridges. The pdfs of the main ozone gain latitude, altitude and time of the trajectories started in the vicinity of the jet stream (Figure 10, black line) look similar to the pdfs of the region west of the high pressure ridges, but show a large variablity among the three weather patterns W1, W4, and W5 (not shown). For example in W4 the jet stream is split

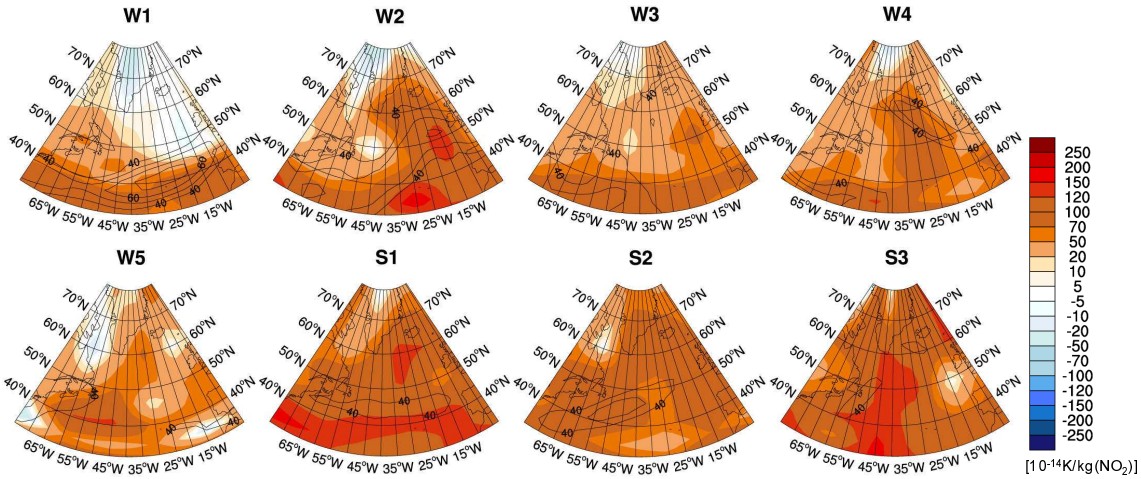

**Figure 11.** Climate change functions of aviation induced total $NO_x$ ($O_3$ + $CH_4$ + PMO) at 250 hPa in $10^{-14}$K/kg($NO_2$) for 8 representative weather situations and an emission at 12 UTC. The isolines show the windspeed in m/s (>40 m/s) at 250 hPa. CCFs for other pressure levels and times investigated in this study are presented in the supplement.

(Figure 2), which leads to a bi-modal distribution (15°N and 35°N) of the main ozone gain latitude, whereas in W1, which is
characterized by a strong zonal jet stream the pdf is unimodal and very narrow with a peak at 30°N. However, in both situations
(W1 and W4) most air parcels released within the jet are transported to lower altitudes within the first two weeks after emission.
This indicates that, similar to the findings for high pressure systems, these air parcels are quickly transported into regions of
high chemical efficiency, resulting in a high climate impact. This again emphasizes the importance of analyzing location and
weather dependent aviation effects, and at the same time supports the potential of finding similarities between corresponding
weather patterns.

  Figure 11 shows the combined CCFs induced by nitrogen oxide ($NO_x$) emissions from aviation. Aviation emissions of $NO_x$
cause an increase of $O_3$, a decrease of $CH_4$ and a methane-induced decrease of $O_3$ (PMO), in the following we denote these
combined effects as total $NO_x$ effect. The total $NO_x$ effect is a combination of increased and reduced warming climate effects
of similar magnitude. Depending on the size of the individual effects, the total $NO_x$-CCF may be either positive or negative.
Similarities between the pattern of $NO_x$-CCF and the weather pattern (Figure 2) are found, with high positive values in the
area of the jet stream and in the area of high pressure ridges, and low or even negative CCFs at high latitudes and in the area
of low pressure troughs. The variability of the pattern is somewhat less pronounced than for $O_3$. Total $NO_x$ effects show a
minimum at ~250 hPa, increasing towards higher and lower altitudes, with a higher (more warming) impact in summer than
in winter (see supplement). An emission release at different times of the day (6, 12, 18 UTC) was found to be negligible and
almost identical CCF results were obtained for $O_3$ and total $NO_x$.

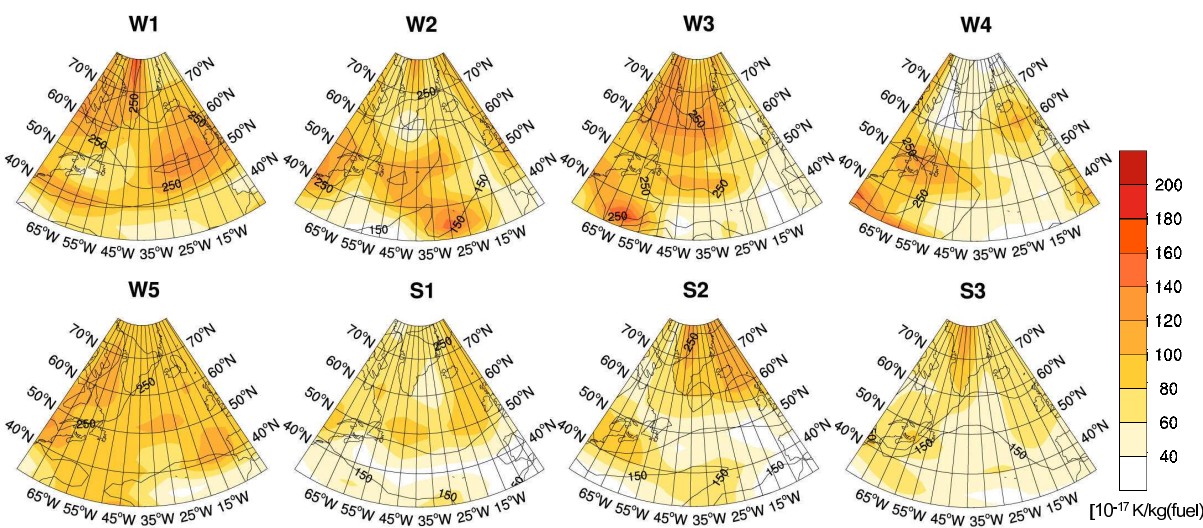

**Figure 12.** $H_2O$ climate change functions for the 8 individual weather situations and an emission at 12 UTC at 250 hPa. CCFs for other pressure levels and times investigated in this study are presented in the supplement. The isolines show the height of the tropopause in hPa.

### 4.3 Water vapour effects

Figure 12 shows the CCFs for $H_2O$ in terms of ATR20 for all weather situations exemplarily at 250 hPa for an emission time of 12 UTC. For this emission altitude (250 hPa) the climate impact varies by approximately one order of magnitude. The variability shows a pattern which clearly reflects the weather situations shown in Figure 2. Where the high pressure ridges
induce a higher tropopause altitude (in W2, W3, W4, S3), the emitted $H_2O$ will rain out more quickly, thus having a shorter residence time, which leads to a lower climate impact of $H_2O$ emissions compared to the regions east or west of the high pressure ridge. Contrary, low pressure troughs correspond to a lower tropopause, thus emissions are released closer to or even in the lowermost stratosphere, where they have a longer residence time and thus a higher climate impact. In summer (S1, S2, S3), the $H_2O$-CCF is considerably smaller than in the winter weather patterns, because of a higher tropopause height, stronger
convective overturning, and thus shorter $H_2O$ lifetimes. In general, the distance of the emission altitude to the actual tropopause largely controls the climate impact of $H_2O$ emissions, as will be discussed in Section 5. Generally, $H_2O$ CCFs increase with height, and are higher in winter than in summer (see supplement). The time of emission release (6, 12, 18 UTC) was found to be negligible and almost identical CCF results were obtained for $H_2O$.

### 5 Discussion

In the previous sections we have shown that there exist large weather related differences of non-$CO_2$ aviation climate effects. We could identify systematic weather related similarities. Regarding contrail-cirrus, we found enhanced potential coverage in

areas where largescale lifting of air masses occurs. Close to the jet stream low potential coverage was observed. These general findings are supported by the study of Irvine et al. (2012), who analysed the distribution of ice-supersaturated regions (ISSRs) in similar weather situations within 20 years of ERA Interim Data (European Centre for Medium-Range Weather Forecasts RE-analysis Interim Data, Dee et al. (2011)). The exact manifestation of potential contrail coverage varies with the characteristic features of the weather situation. In general, our findings correspond well with the potential contrail cirrus coverage simulated by Burkhardt et al. (2008), given that their numbers (17%–21%) for 230–275 hPa comprise only 30–60°N. Regarding contrail lifetimes, our mean lifetimes of 3.5±5.3 h agree well with the estimates of Gierens and Vazquez-Navarro (2018), who determined the complete lifetime of persistent contrails to be 3.7±2.8 h by applying an automatic tracking algorithm in combination with statistical methods to one year of Meteosat-SEVIRI data over Europe and the North-Atlantic. In their example, 80% of contrails had a lifetime smaller than 5 h and 5% lived longer than 10 h. Figure 13 compares the cumulative probability density function of lifetimes of both studies, illustrating that in the present study, a comparably larger fraction of contrails has lifetimes below 3 h. Both the initial as well as the final stages of contrail lifetimes could only be estimated by Gierens and Vazquez-Navarro (2018), as these stages are difficult to observe by satellite platforms. Ice water contents and optical depths for contrails of the REACT4C study were already presented in Grewe et al. (2014a), and were found to compare well with other studies, e.g. Kärcher et al. (2009); Frömming et al. (2011); Voigt et al. (2011).

Whether the climate impact of contrails and contrail cirrus is warming or cooling in the respective situation is complex and involves detailed knowledge about e.g. contrail optical properties, contrail lifetimes, solar zenith angle, ambient cloud coverage and surface properties below the contrail (e.g. Schumann et al., 2012). In General, enhanced climate impact of contrail-cirrus (irrespective if positive or negative) was detected south of the jet stream, in the vicinity of Greenland and in areas with strong meridional transport, whereas comparably low contrail-cirrus climate impact is found close to the jet stream. More detailed and smaller-scale structures of contrail formation areas cannot be resolved with the present model resolution. A higher spatial and temporal resolution of time-regions and of the underlying atmospheric conditions would be favourable for future studies on contrail-CCFs.

Regarding $O_3$- and total $NO_x$-CCFs, we identified high positive values in the area of the jet stream and in the area of high pressure ridges, whereas low or even negative CCFs were found at high latitudes and in the area of low pressure systems. In general, the climate impact is higher in summer than in winter, because of reduced photochemistry due to missing sunlight in winter. These findings are in qualitative agreement with earlier climatological studies showing higher responses for $NO_x$ emissions at low latitudes and lower or even negative effects at high latitudes (e.g. Berntsen et al., 2005; Köhler et al., 2008; Grewe and Stenke, 2008; Köhler et al., 2013) and comparable seasonal effects (e.g. Gauss et al., 2006). A newer study by Lund et al. (2017) also found largest effects of aviation $NO_x$ for emissions in the tropics and subtropics and smaller effects for emissions at midlatitudes. They state that the sign and magnitude of total $NO_x$ climate effects in such studies depend strongly on the chosen metric and time horizon. Furthermore there exist large inter-model differences in the change in methane lifetime per unit ozone change as discussed by e.g. Köhler et al. (2008); Myhre et al. (2011). In comparison with studies regarding annual averages, the present study enables a representation of more detailed transport pathways, characteristic of synoptic

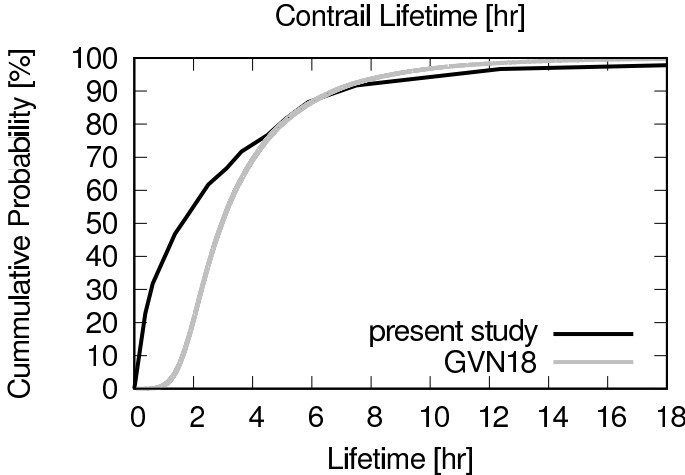

**Figure 13.** Cumulative probability of contrail lifetimes for all weather situations in comparison to Gierens and Vazquez-Navarro (2018) (=GVN18).

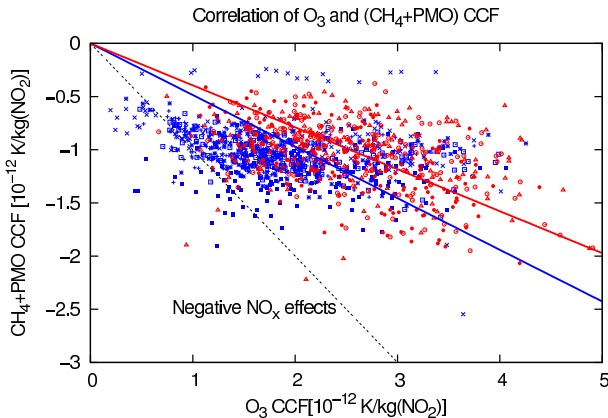

**Figure 14.** Correlation of $O_3$-CCFs and the $CH_4$- and PMO-CCFs. The dashed line indicates the transition, where the total $NO_x$ effect becomes negative. The symbols indicate the different weather patterns (blue = winter, red = summer). The blue and red lines show a linear fit for winter and summer weather patterns, respectively.

weather patterns. The knowledge that can be gained from this study, focusing on synoptic weather patterns, is that not only the emission region is significant but even more, the tranport pathways from that emission region towards other chemical regimes. A correlation between the climatological response of $O_3$ and $CH_4$ to $NO_x$ emissions has been shown in many studies (e.g. Lee et al., 2010; Holmes et al., 2013; Köhler et al., 2013). Our data demonstrate that this relation continues when regarding individual weather situations. Figure 14 shows the $O_3$-CCFs and the combined $CH_4$-PMO-CCFs for all weather situations. A


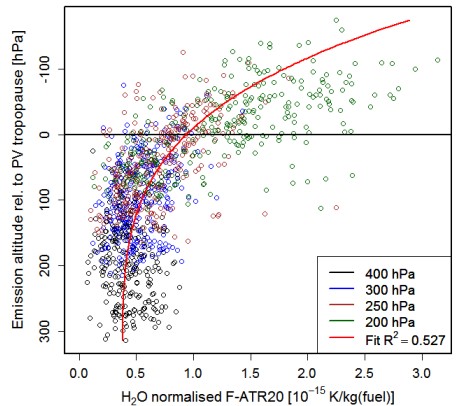

**Figure 15.** Correlation of water vapour climate change functions [K/kg(fuel)] with emission altitude difference relative to the actual tropopause. Pressure levels of the various emission altitudes are distinguished by different colours. A fit function is indicated by the red line.

clear correlation is found, indicating that actual weather influences both effects in a similar way, with large or small positive values of $O_3$-CCFs correlated with large or small negative $CH_4$- and PMO-CCFs. However, the variability of the $O_3$-CCFs ($\pm$ 1.5 $10^{-12}$ K/kg($NO_2$) is about a factor of 3 larger than the combined $CH_4$/PMO variability ($\pm$ 0.5 $10^{-12}$ K/kg($NO_2$).

The $H_2O$-CCFs were also found to be closely related to the actual weather pattern. In regions with higher tropopause altitudes the emitted water vapour has a shorter atmospheric residence time and thus a lower climate impact, whereas in regions with lower tropopause height, the emitted water vapour has a longer residence time and a higher climate impact. In the summer situations, the $H_2O$-CCFs are generally smaller than in winter because of enhanced convective activity (larger vertical mixing) and subsequent rainout of $H_2O$ and a generally higher tropopause height. Figure 15 shows the correlation of $H_2O$-CCFs to the

emission altitude relative to the tropopause. The $H_2O$ emission from 1 kg fuel occuring below the tropopause, yields a warming of approximately $0.5 \cdot 10^{-15}$ K, whereas the same emission above the tropopause leads to a warming of around 1 to $3 \cdot 10^{-15}$ K. In general, the distance of the emission altitude to the actual tropopause largely controls the climate impact of $H_2O$ emissions. The higher the water vapour emissions are released (relative to the tropopause), the longer it takes until this water vapour enters the troposphere and will eventually be rained out, i.e. the longer is its residence time. These findings are supported by earlier

studies regarding the climate effect of water vapour emissions from aviation in a climatological sense (e.g. Grewe and Stenke, 2008; Fichter, 2009; Frömming et al., 2012; Wilcox et al., 2012).

   Our findings are in agreement with earlier studies, which investigated the altitude and latitude dependency of annual mean or seasonal non-$CO_2$ aviation effects (e.g. Gauss et al., 2006; Köhler et al., 2008; Grewe and Stenke, 2008; Fichter, 2009; Frömming et al., 2012; Köhler et al., 2013). Furthermore, as far as comparable, our findings are also in qualitative agreement

with studies which investigated the avoidance of contrails (e.g. Mannstein et al., 2005; Sridhar et al., 2011; Chen et al., 2012; Irvine et al., 2014; Zou et al., 2016; Hartjes et al., 2016; Yin et al., 2018a), although the present study does not optimize tra-

jectories but is only setting the scene. These previous studies indicated a strong reduction potential but were only valid for the actual situation and could not be transferred to other situations.

The weather situations which were selected in our study occured in the months December to February and June and July.
Although our findings might be transferable to other seasons, future studies should look at special features, which might occur in other months.

In the present study, we have explicitly excluded the direct and indirect climate impact of aviation aerosols. The status of knowledge on indirect aerosol effects is not considered being mature enough to be included in such a study. This will be covered in future projects.

It is essential to note that uncertainties are associated with individual climate change functions presented in this study. However, for the application of these data in terms of optimisation of flight trajectories not the exact value of climate impact is crucial, but the relation of the individual components and their spatial and temporal variability. Grewe et al. (2014b) performed a detailed study on the sensitivity of routing changes with respect to uncertainties and potential errors in the climate change functions. Their sensitivity studies approximately cover the range of uncertainty of individual aviation climate effects as specified by
Lee et al. (2010) and they investigated an increased variability of the individual atmospheric responses calculated. This had an effect on the weighting between the climate impact from NOx and contrail-cirrus. Grewe et al. (2014b) found differences in the reduction potentials but similar optimal routes. These sensitivity studies indicate a stable response in the shape of the cost benefit analyses and the way air traffic is routed for climate impact.

The CCFs of the individual species show the sensitivity of the atmosphere to non-$CO_2$ aviation emissions. If flight trajectories
were rerouted to reduce climate impact by avoiding the most sensitive regions, possible tradeoffs between individual species need to be considered. With the present study, these tradeoffs can be estimated in a consistent way as the effects of all species are represented by means of a consistent metric. For the first time, a comprehensive data set is available for various species, pressure levels, emission times and a multitude of weather situations. During optimization, the characteristic effects of all species can be equally compared or individually be assigned with different weights. As a first step for a rough comparison, all
CCFs are converted to K/kg(fuel). We multiply the contrail-cirrus CCF by a typical specific range value for transatlantic flights of 0.16 km/kg(fuel) (Graver and Rutherford (2018) and personal communication F. Linke, DLR). Similarly, the total $NO_x$-CCF is converted using a typical emissions index of $NO_x$ of 13 g($NO_2$)/kg(fuel) (Penner et al., 1999). Figure 16 shows the merged CCFs of contrail-cirrus, total $NO_x$ and $H_2O$ exemplarily for weather situation S2 at 250 hPa and 12 UTC. When evaluating the individual components of the merged CCFs, it is clearly revealed, that contrail-cirrus and $O_3$-CCFs are the dominating
non-$CO_2$ effects. A hypothetical climate optimized transatlantic flight (which will stay on this pressure level for simplification) would certainly try to avoid the area with high positive CCFs in the western Atlantic around 40°N, which is due to warming contrail-cirrus and warming total $NO_x$ effects. Further this flight trajectory will probably find a compromise between avoiding long distances through enhanced climate warming areas and at the same time avoiding long detours as these would induce a penalty with respect to $CO_2$-CCF. However, situations are conceivable, where extensive areas with cooling contrails occur
(similar to the negative CCF area in the central Atlantic), which flight trajectories might purposely seek during optimization to minimize their overall climate impact. We emphasize, that this is a very simplified example to illustrate the concept. The op-

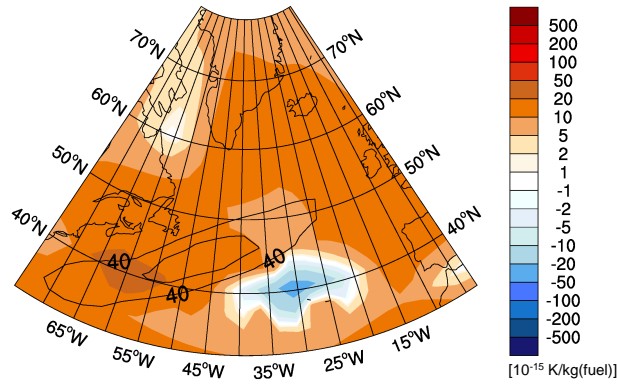

**Figure 16.** Merged climate change functions in $10^{-15}$ K/kg(fuel) of aviation induced total $NO_x$, contrail-cirrus and $H_2O$ at 250 hPa exemplarily for weather situation S2 for an emission at 12 UTC.

timization of weather dependent flight trajectories with respect to minimum climate impact is much more complex. However, such an optimization goes clearly beyond the scope of the present study. Nonetheless, the data from the present study are a comprehensive and valuable basis for weather dependent flight trajectory optimization with minimum climate impact. Some
of the studies, based on the present data have already been published (e.g. Grewe et al., 2014b; Niklass et al., 2017; Yin et al., 2018b; Yamashita et al., 2020), others are in preparation.

Common features of the non-$CO_2$ CCFs facilitate the usage of our results for the development of more generalized algorithmic climate change functions (aCCFs). If CCFs were intended to be used for actual climate optimal flight planning, it would be necessary to predict the sign and magnitude of individual CCFs for the actual weather situation. Due to excessive use of com-
puting time, it is not possible to calculate CCFs in detail for any actual situation. A procedure would be necessary to bypass detailed simulations. On the basis of specific CCFs from the present study, more generic Climate Change Functions, so called algorithmic Climate Change Functions (aCCFs) were developed. These algorithms facilitate the prediction of CCFs by means of instantaneous meteorological data from weather forecasts without the necessity of computationally extensive recalculation of CCFs by means of chemistry-climate model simulations. This was the aim of the related studies by Van Manen and Grewe
(2019) and Yin et al. (2020). A number of assumptions and simplifications were necessary for such an approach. Nevertheless, algorithms have been devised for the prediction of $O_3$-, $CH_4$-, $H_2O$- and contrail-cirrus CCFs. These aCCFs would facilitate weather related climate optimized planning of flight trajectories for any weather situation. Such a modelling study has been performed by e.g. Yamashita et al. (2020), who implemented the aCCFs in a flight planning tool, to optimize flight trajectories with regard to various objective functions.


## 6   Summary and Conclusions

We investigated the influence of different weather situations on the climate impact of non-$CO_2$ aviation emissions. Our results are 4D-climate change functions, which describe the climate impact of local emissions of $NO_x$ and $H_2O$, affecting the formation of contrail-cirrus and the mixing ratios of the greenhouse gases $O_3$, $CH_4$ and $H_2O$. We studied the impact of local emissions for eight different representative weather situations and for three points in time per day, resolving the temporal evolution of the weather system. The main objective was to derive systematic relationships between the emission location, the prevailing weather situation during emission and the resulting aviation climate impact. The model configuration and methodology to generate spatially and temporally resolved information on the sensitivity of the atmosphere to local aviation emissions, which has been employed in this research paper, is, to our knowledge, unique.

For all non-$CO_2$ species included in this study, we found distinct weather related differences in their associated CCFs. We found an enhanced significance of the position of emission release in relation to high pressure systems, to the jet stream, to the altitude of the tropopause, and to polar night. Regarding chemical effects of aviation $NO_x$ emissions, we find, that not only the emission region is relevant, in fact, the main driver for enhanced climate impact sensitivity are transport pathways of emissions within the first week(s) after emission. If emissions are released west of a high pressure system, they stay or are transported to high latitudes, resulting in minor $O_3$ production. If emissions are released in a high pressure system, they are transported towards lower altitudes and latitudes and experience strong photochemical $O_3$ production, causing enhanced climate impact. The transport pathways are adequately represented in the present study and weather patterns compare very well with the classification using ERA-Interim data (Irvine et al., 2013). The climate impact of $H_2O$ emissions is largely controlled by the distance to the tropopause, with emissions released close to the tropopause or even in the lowermost stratosphere causing the strongest climate impact. Diurnal effects are negligible for aviation $NO_x$ and $H_2O$ emissions. Regarding contrail-cirrus CCFs the results are too diverse to be summarized easily. The main factor for contrail-cirrus climate impact are ambient conditions during contrail formation and some hours after. The diurnal variation of insolation is crucial in combination with the lifetime of contrails as it has the potential to change the sign of contrail climate impact. Reproducing a higher degree of detail regarding the small-scale structure of contrails and contrail-cirrus and their temporal variation could be improved in future studies through enhanced spatial and temporal resolution. However, whether a smaller scale structure of contrail-cirrus CCFs would have an impact on flight trajectory optimization has not yet been investigated.

The results of this study represent a comprehensive dataset for studies aiming at weather dependent flight trajectory optimization reducing total climate impact. The dominating non-$CO_2$ effects were found to be contrail-cirrus and impacts induced by $NO_x$ emissions on average, however, this might deviate temporally and regionally. For an implementation of climate change functions in actual flight planning it would be necessary to accurately predict the sign and magnitude of individual CCFs for the actual weather situation. This can possibly be persued by means of more generic aCCFs, which facilitate the prediction of CCFs by means of instantaneous meteorological data (e.g. Matthes et al., 2017). These aCCFs have to be verified and first verification results for the $O_3$-aCCFs are promising (Yin et al., 2018b). However, to improve the quality of these predictions, more knowledge has to be gained, particularly with repect to the transition of warming to cooling climate effects from contrail-

cirrus and total $NO_x$ impacts. Further evaluation and quantitative estimates on uncertainties require additional comprehensive climate-chemistry simulations. Furthermore, better understanding of the tradeoffs between different effects (e.g. transport versus chemistry) or different species is essential. It might also be useful to focus on evaluating what might be the most promising regions to bypass, in other words, where total climate impact is highest and easiest to avoid or at lowest additional cost. The CCFs presented in this study represent the indispensable data basis for climate-optimized flight planning. The potential im-

plementation of such an approach faces several challenges, a roadmap how to overcome these is elaborated in Grewe et al. (2017).

*Code and data availability.* The Modular Earth Submodel System (MESSy) is continuously further developed and applied by a consortium of institutions. The usage of MESSy and access to the source code is licenced to all affiliates of institutions which are members of the MESSy Consortium. Institutions can become a member of the MESSy Consortium by signing the MESSy Memorandum of Understanding.

More information can be found on the MESSy Consortium Website (http://www.messy-interface.org). The results presented here have been obtained with a modified MESSy version 2.42, these modifications became part of the official release of MESSy version 2.50. The results presented in this work are archived at DKRZ and are available on request. A part of the data is available via the REACT4C project home page (https://www.react4c.eu/).

**Appendix A: Calculation of the main ozone gain latitude, altitude, and time**

The main ozone gain latitude ($\phi_j$) of an emission location (identified with the index $j$) is defined as the mean latitude at which the air parcel trajectories experience most of the ozone increase. Accordingly, the main ozone altitude and time are defined as the mean altitude and time at which the air parcel trajectories experience most of the ozone increase, respectively. In the following, we exemplarily define how the ozone gain latitude is derived, the other quantities are obtained by replacing latitude by altitude and time, respectively. The air parcel trajectories (identified with the index $i$) will contribute different shares to the

total ozone gain latitude. The ozone gain ($O_3^{Gain_i}(t)$) along an air parcel trajectory is defined as the increase in $O_3$ from the previous to the current time step $t$ (for a decrease in $O_3$, the ozone gain is set to 0). The contribution $A_{j,i}$ of a single trajectory $i$ to the latitude of the main ozone gain (=ozone gain latitude ) for the emission location $j$ is given by

$$A_{j,i} = \int \frac{O_3^{Gain_i}(t) \cdot \phi_i(t)}{\sum_{i=1}^{50} \int O_3^{Gain_i}(t)dt} dt, \tag{A1}$$

where $\phi_i(t)$ is the latitude of the trajectory $i$ at time $t$. By taking the sum of the contributions $A_{j,i}$ of all trajectories $i$ starting

at emission location $j$, the latitude of the main ozone gain is

$$\phi_j = \sum_{i=1}^{50} A_{j,i}. \tag{A2}$$

The weights ($w_{j,i}$) for each trajectory to the ozone gain latitude are calculated by combining Equation (A1) and (A2):

$$w_{j,i} = A_{j,i}/\phi_j. \tag{A3}$$

**Table A1.** Overview of all considered locations for the weighted probability density functions.

| High Pressure Ridge | | West of High Pressure Ridge | | Max. $O_3$-CCF near Jet Stream | | |
| W3 | W4 | W3 | W4 | W1 | W4 | W5 |
| --- | --- | --- | --- | --- | --- | --- |
| $40°N, 15°W$ | $35°N, 30°W$ | $40°N, 30°W$ | $35°N, 75°W$ | $35°N, 60°W$ | $30°N, 60°W$ | $40°N, 75°W$ |
| $45°N, 15°W$ | $40°N, 30°W$ | $45°N, 30°W$ | $40°N, 60°W$ | $35°N, 45°W$ | $30°N, 45°W$ | $40°N, 60°W$ |
| $50°N, 15°W$ | $45°N, 30°W$ | $50°N, 30°W$ | $45°N, 45°W$ | $35°N, 15°W$ | $30°N, 30°W$ | $40°N, 45°W$ |

A similar procedure is used to calculate the latitude of ozone gain for each single trajectory $\phi_{j,i}$:

$$\phi_{j,i} = \int \frac{O_3^{Gain_i}(t) \cdot \phi_i(t)}{\int O_3^{Gain_i}(t)dt}dt. \tag{A4}$$

The main difference between $\phi_j$ and $\phi_{j,i}$ is the weighting of the latitude. For $\phi_{j,i}$ the ozone gain of a single trajectory is taken into account, whereas for $\phi_j$ the ozone gain of all trajectories started at the emission region $j$ is taken into account. Equation (A3) and (A4) define a data set containing the contribution and the latitudinal location of the main ozone gain for each trajectory. Based on these data a weighted probability density function (PDF) is derived in Eq. (A5). For a bin size of $\Delta\phi$, a center $\phi$ of this bin, and $n$ air parcel trajectories ($i = 1, ..., n$), which have their main ozone gain $\phi_{j,i}$ in this bin, the PDF is:

$$pdf(\phi) = \frac{\sum_{j=1}^{n} w_{j,i}}{\sum w_{j,i} \cdot \Delta\phi}. \tag{A5}$$

The sampling size of this PDF would be rather small, if only a single emission location was taken into account (50 trajectories). In order to enhance the data basis, trajectories from various emission grid points are sampled for different meteorological features (High Pressure Ridge, West of High Pressure Ridge, and Near Jet Stream). In case 1 ("High Pressure Ridge"), the maximum of the $O_3$-CCFs is analyzed for W3 and W4 which are both in the region of a high pressure ridge. In case 2 ("West of High Pressure Ridge"), the same weather situations (W3 and W4) are analyzed, but the emission locations evaluated lie further west compared to the points of case 1. In this case, the $O_3$-CCFs are significantly lower. In both cases the same emission latitudes are taken into account but different longitudes. The last case ("Near Jet Stream") considers the location of high $O_3$-CCFs in the vicinity of the jet stream. For this case weather patterns W1, W4 and W5 are analyzed. A summary of all emission locations taken into account is given in Table A1. Results are discussed in section 4.2.

*Author contributions.* CF, VG and SM designed the study. CF and PJ developed the relevant EMAC submodels with input from VG and SB. CF and SB performed the EMAC simulations. AH calculated the CCFs from the EMAC output with contributions from CF and VG. CF analysed the data with contributions from VG, SR, JVM. CF prepared the manuscript with contributions from all co-authors.

*Competing interests.* Author PJ is a member of the editorial board of the journal.

*Acknowledgements.*  This work was supported by the European Union FP7 Project REACT4C (React for climate: http://www.react4c.eu/). Computational resources were made available by the German Climate Computing Center (DKRZ) through support from the German Federal Ministry of Education and Research (BMBF) and by the Leibniz-Rechenzentrum (LRZ). We gratefully acknowledge helpful discussions with Emma A. Klingaman (formerly Irvine) (University Reading), Ruben R. de Leon (Manchester Metropolitan University), Keith Shine (University Reading) and Katrin Dahlmann (DLR). We thank Helmut Ziereis (DLR-internal review) for many helpful comments.

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
