# Peer review of "Influence of weather situation on non-CO2 aviation climate effects: The REACT4C Climate Change Functions"

_Atmospheric Chemistry and Physics, 2020_

## Referee Comment (RC1) · Anonymous Referee #1 · 21 Dec 2020

Review of Frömming et al.: Influence of the actual weather situation on non-CO2 aviation climate effects: The REACT4C Climate Change Functions

SUMMARY:

The study presented in this manuscript investigates how typical weather situations in the North Atlantic region affects the most important non-CO2 climate change mechanisms from aviation emissions. The North Atlantic flight corridor, which connects Europe and North America, is one of the busiest regions for commercial air travel and the authors give very good reasons for their geographical choice. The authors have a track record of publications in this field and demonstrate awareness of the published lit-

erature, even though they appear to focus more on publications by European research groups. The model chosen for this study has been used previously to assess the impact of non-CO2 aviation emissions in a climate context. In summary I feel this is a good manuscript about a worthwhile study which has been carried out with diligence. It demonstrates good scientific significance and quality and is presented in a focused manner. Therefore, I would like to congratulate the authors to this manuscript. Having said this, I have several concerns that I would like the authors to address before I recommend accepting the manuscript for publication. While there are many good aspects about this manuscript, the remaining part of my review will focus on the areas where I think improvements are necessary.

GENERAL CONCERNS:

This paper is one of several publications based on work from the REACT4C project, hence there are many references to companion papers which describe partly the model and methodologies. It would be helpful if the authors can describe very clearly the novelty and individual contribution presented in this manuscript. The authors state that apart from a Grewe et al. 2014 pilot study this is the first assessment of the impact of weather situations on aircraft climate impacts. I am making this point because, for instance, a subset of the authors of this manuscript have also authored the Rosanka et al. ACP companion paper where there seems to be partly an overlap with scientific aims. The Rosanka paper is however only mentioned late and rather cursory. Further, the Irvine et al. 2013 study also has been a precursor study for this work. It would be good to explain more clearly and early in the manuscript how this study distinguishes itself from the other studies and how the REACT4C publications relate to each other.

The authors correctly state that previous publications focused on aircraft emission climate impacts from an annual average/climatological perspective. An important reason for this is that global chemistry-climate models (at least those comprising a reasonably detailed treatment of tropospheric and stratospheric chemistry) are computationally so expensive that the higher grid resolutions which are typically employed to study synoptic scale weather patterns are beyond the reach of these models. In the study presented here, the authors have employed a model with a quadratic Gaussian grid at T42 which results in grid point spacing greater than 300 km at the Equator. The spacing will become a bit smaller at mid-latitudes where the focus of their study lies, but it still is substantially coarser than grid resolutions used for synoptic scale considerations. Naturally, the authors are limited by the above-mentioned computational constraints. Do they consider the grid resolution as a constraint for the validity of their results at all, given that they are focusing on the impact of synoptic scale weather patterns rather than multi-annual averages? Would they expect their findings to be sensitive to grid resolution? This has not been mentioned at all and I would think this could be a considerable caveat. I would like these considerations included in the discussion, given that emissions from aircraft flight tracks, contrail formation, and many synoptic processes are sub-grid scale processes for the grid resolution that is used here.

The technical description how the experiments are carried out remains unclear and needs to be described better. In Section 2 the components of the modelling system are listed diligently by referring to large number of citations, however how the 280 experiments relate to the 4032 perturbation events, the particulars of the trajectories that are released at the perturbation points, and how all this leads to the results shown in the figures remains to some extent a mystery to me. There is much more clarity needed.

SPECIFIC COMMENTS:

l. 42: Lee et al 2010 remains an important paper but it is now more than 10 years old. Perhaps add Brasseur et al 2016 or other more recent review/assessment papers?

l.71: Correct typo 'respect'. What is meant by "user defined" penalty factors – please explain this a bit clearer.

l. 89: In my opinion the companion paper Rosanka et al., ACP, 2020 should be mentioned already here and not only in l. 323, as their findings appear to result also from

actual weather patterns (related to the same project?)

l. 107: The entire Section 2 forms a dense amount of text which is difficult to digest as one block. I suggest structuring it more clearly by breaking into subsections. I would suggest at least a section describing the model components, and a further section describing the set up used for the experiments and how the model outputs are obtained (combination of Eulerian and Lagrangian approach?). A dedicated description of how the experiments were conducted is missing, instead it seems to be scattered over the remaining parts of the manuscript. Please draw this together in a clearly reproducible experiment description.

l. 131: Why is 00 UTC not part of the emission times? On what criteria have the authors chosen the magnitude of the NOx and H2O emission perturbations, are they in any way related to an emissions inventory? Particularly the H2O value looks oddly precise. Also, are these perturbations in addition to the existing aircraft emissions at the grid point location?

Table 1: Thinking about the background conditions at these grid point locations: How do they overlap with the present flight routing/aircraft emissions distribution? Are they all located inside the main North Atlantic flight corridor?

l. 207: In the interest of understanding, clarity and reproducibility, it is not clear how the perturbations were applied. Obviously, this study consisted of a substantial number of model experiments (as indicated by the total CPU hours), however the 280 experiments remain still an order of magnitude below the 4032 perturbation events. How were they grouped together? How many perturbations were applied during one experiment? How many experiments represent one weather situation? Further, after a detailed listing of the model's abilities and sub-components (with citation) in Section 2 of the manuscript, there is a lack of technical detail how the experiments were conducted. Were they all initialised from a common set of initial conditions, perhaps one for each weather situation? How long was the model spun up for prior to the 90 days of the output that
was taken into consideration?

Figures 2 and 3: It is not clear how these findings relate to the experiments: Is this the additive result of all the perturbations combined? How do you obtain from perturbations in individual grid cells, which are propagated through trajectories, a regional map of potential contrail coverage in %? This should become clear through an improved experiment description in Section 2. Similarly, this will also help interpreting Figs 7, 10 and 11.

While the method of compiling the information needs better explaining, I would like to point out that I find the information conveyed by these maps of great value and I welcome this method of visualising the results in general (and I am glad that figure labelling has a reasonable, legible size!)

Section 4.2: My concern here stems again mainly from a lack of understanding how to interpret the figures. While contrails are formed almost instantly in the wake of the aircraft, impacts from NOx emissions have a longer lifetime and the map representation in Figs 7 and 10 only makes sense if it indicates the location of the emissions release (even though the climate change impact might be felt elsewhere). Can the authors confirm this?

l. 379: It would be interesting to know the author's view on why the jet stream region would yield generally more positive values, seeing that otherwise subtropical high-pressure belts exhibit positive values.

l. 382: Here the authors compare their findings with climatological studies that were partly carried out with even lower grid resolution (e.g. T21). Previously the authors have shown the importance of the pathway of NOx emissions which would be better resolved with higher model grids. To what extent do the authors believe a T42 representation of short-lived weather situations offers a fair comparison to climatological studies? While I can understand that it can instil confidence to have findings agree with previous studies, I would like to be reassured that the comparison is on fair terms?

How important do the authors consider grid resolution in their study? What is the minimum resolution required to resolve the weather situations sufficiently to represent the emissions transport on non-climatological time scales?

l. 413: explicitly (typo)

Conclusions: The first paragraph is fairly high level, only telling me that jet stream location, tropopause height, high pressure systems and polar night are all having an influence, but this is not further elaborated. It would be nice if the main findings or messages could be brought more clearly across. Is there a pattern that the authors recognise? If the results are too diverse to summarise then perhaps it should be stated here.

l. 469 or l. 479: As previously mentioned, the conclusions here need to acknowledge the limits in grid resolution and what impact the authors believe this might have on their findings.

---

## Referee Comment (RC2) · Anonymous Referee #2 · 22 Jan 2021

The paper address non-CO2 climate impact of aviation, quantifies and presents climatology of these effects for the North Atlantic and discuss possibilities for mitigation of these impacts through alternative routing. This is a very complex scientific question relevant for ACP, also very topical as the non-CO2 climate impacts, if unresolved, makes it difficult for the aviation sector to achieve the Paris agreement targets. The paper present results of model simulations where the local aircraft emissions are followed in a global ECHAM5/MESSy Atmospheric Chemistry model system on Lagrangian trajectories. There are several novel modules developed within the model system associated with these calculations which have been published and are cited as accompanying papers, the most important Grewe et al. (2014) and Rosanka et al. (2020). While Grewe

et al. (2014) presents results for 1 case (weather situation), this paper presents results for a number of situations under winter and summer season which allows for more general conclusions based on thousands of simulated trajectories and their complex analysis. These papers represent an impressive piece of work which gives a new insight in climatology of non-CO2 climate effects of aviation and I would like to congratulate the authors to this achievement. Before recommending the paper for publication, I would however like to draw their attention to the following issues: The methods and assumptions in the paper are extremely complex and described in many cases rather briefly with references to other papers. This makes it rather difficult to follow and assess the methodology. I would recommend including more comprehensive and systematic description of the methodology which would give clear idea about how the crucial processes are treated in the model simulations performed for this paper. Even when going to the references I could not follow how the chemistry of the 'multitudes of background trajectories' which the local emission trajectory is mixed with is calculated, recommend explanation of this part in particular. After reading the papers describing the tagging mechanism for quantification of impact of studied emissions, I would like to ask if the non-linear plume effects on ozone formation from NOx emissions, or rather on NOx removal, are considered in some way and, as this has been subject of scientific discussion under quite a long time, what impact these effects have or could have (in case they were not considered). Specific comments: Fig. 2 – Potential contrail coverage for the representative weather situations – what is the figure showing – mean over certain time period of each weather situation? Supplement – Caption or some explanation to the figures is missing

---

## Author Comment (AC1) · 5 Mar 2021

Review of Frömming et al.: Influence of the actual weather situation on non-CO2 aviation climate effects: The REACT4C Climate Change Functions

SUMMARY:
The study presented in this manuscript investigates how typical weather situations in the North Atlantic region affects the most important non-CO2 climate change mechanisms from aviation emissions. The North Atlantic flight corridor, which connects Europe and North America, is one of the busiest regions for commercial air travel and the authors give very good reasons for their geographical choice. The authors have a track record of publications in this field and demonstrate awareness of the published literature, even though they appear to focus more on publications by European research groups. The model chosen for this study has been used previously to assess the impact of non-CO2 aviation emissions in a climate context. In summary I feel this is a good manuscript about a worthwhile study which has been carried out with diligence. It demonstrates good scientific significance and quality and is presented in a focused manner. Therefore, I would like to congratulate the authors to this manuscript. Having said this, I have several concerns that I would like the authors to address before I recommend accepting the manuscript for publication. While there are many good aspects about this manuscript, the remaining part of my review will focus on the areas where I think improvements are necessary.

We thank the anonymous referee for seeing the value of our work and estimating the work as relevant for ACP. We gratefully appreciate his/her time reading our manuscript thoroughly and pointing towards paragraphs which were not clear enough or needed improvement. We amended the manuscript accordingly. Please find our responses (red) to the review points and recommendations (black) below.

GENERAL CONCERNS:

This paper is one of several publications based on work from the REACT4C project, hence there are many references to companion papers which describe partly the model and methodologies. It would be helpful if the authors can describe very clearly the novelty and individual contribution presented in this manuscript. The authors state that apart from a Grewe et al. 2014 pilot study this is the first assessment of the impact of weather situations on aircraft climate impacts. I am making this point because, for instance, a subset of the authors of this manuscript have also authored the Rosanka et al. ACP companion paper where there seems to be partly an overlap with scientific aims. The Rosanka paper is however only mentioned late and rather cursory. Further, the Irvine et al. 2013 study also has been a precursor study for this work. It would be good to explain more clearly and early in the manuscript how this study distinguishes itself from the other studies and how the REACT4C publications relate to each other.

We agree, that this needed clarification and that an overview over the REACT4C related studies might be helpful. We added a paragraph explaining the variety of REACT4C related studies and their particular focus and how the studies relate. Furthermore, we tried to emphasize what is unique in the present study and how it discriminates from the other studies. Please find also a figure below, which schematically explains the overview on studies and their relation (l. 87ff).

[Figure]

The authors correctly state that previous publications focused on aircraft emission climate impacts from an annual average/climatological perspective. An important reason for this is that global chemistry-climate models (at least those comprising a reasonably detailed treatment of tropospheric and stratospheric chemistry) are computationally so expensive that the higher grid resolutions which are typically employed to study synoptic scale weather patterns are beyond the reach of these models. In the study presented here, the authors have employed a model with a quadratic Gaussian grid at T42 which results in grid point spacing greater than 300 km at the Equator. The spacing will become a bit smaller at mid-latitudes where the focus of their study lies, but it still is substantially coarser than grid resolutions used for synoptic scale considerations. Naturally, the authors are limited by the above-mentioned computational constraints. Do they consider the grid resolution as a constraint for the validity of their results at all, given that they are focusing on the impact of synoptic scale weather patterns rather than multi-annual averages? Would they expect their findings to be sensitive to grid resolution? This has not been mentioned at all and I would think this could be a considerable caveat. I would like these considerations included in the discussion, given that emissions from aircraft flight tracks, contrail formation, and many synoptic processes are sub-grid scale processes for the grid resolution that is used here.

Thank you for pointing us toward this concern, we follow your suggestion to include this aspect in our discussion and conclusion (l. 493ff, l. 590ff). The characteristic weather patterns as defined in Irvine et al., 2013, who used 0.7° reanalysis data are represented well in our global simulation with T42 resolution (2.8°), relying on a classification by means of atmospheric indices. Comparing respective typical members from EMAC and ECMWF shows that the overall synoptic structure is well captured on the specific day, while the analysis of frequency of these weather patterns reveals differences between ECMWF and EMAC. Specifically, patterns with high atmospheric NAO and AO indices show a lower probability within EMAC, hence their occurrence being underrepresented. Accordingly, we can conclude from this analysis that our analysis relying on characteristic members represents the synoptic scale meteorological patterns. Hence, chemical effects where the main driver are the transport pathways,

are realistically represented within the resolution employed. Understanding the details of the impacts during episodic situations in our view helps to better understand the multi-annual averages. However, from model comparisons with observations it is well known that the small-scale structures of contrail formation regions are at sub-grid resolution (cloud resolving) and are not well reproduced. Hence, in our study only those contrail effects being part of large (synoptic) scale structures can be represented, which have been detected in satellite observations. Overall, we expect that our estimates on contrail and contrail cirrus formation are sensitive to model resolution. Future studies would benefit from investigating the importance of increased spatial and temporal resolution.

The technical description how the experiments are carried out remains unclear and needs to be described better. In Section 2 the components of the modelling system are listed diligently by referring to large number of citations, however how the 280 experiments relate to the 4032 perturbation events, the particulars of the trajectories that are released at the perturbation points, and how all this leads to the results shown in the figures remains to some extent a mystery to me. There is much more clarity needed.

Thank you for pointing this out. We have added extensive and much more detailed description of the basic methodology (2.1), the calculation of contrail (2.3) and chemical effects (2.2), the calculation of radiative forcing and climate change functions (2.4) for the time-regions. In addition, we implemented a Section (2.5) describing the experimental set up.

SPECIFIC COMMENTS:

l. 42: Lee et al 2010 remains an important paper but it is now more than 10 years old. Perhaps add Brasseur et al 2016 or other more recent review/assessment papers?

This is correct. We added two more recent assessment papers (Brasseur et al., 2016 and Lee et al., 2021) and updated the global numbers in the introduction. At the time we submitted the manuscript, the new Lee et al. (2021) was not published yet, but as it is now, of course we consider it as an important reference, so is the Brasseur et al., 2016 paper.

l.71: Correct typo 'respect'.  What is meant by "user defined" penalty factors – please explain this a bit clearer.

We rephrased the sentence and hope it is now clearer what was meant. (l. 72ff)

l. 89: In my opinion the companion paper Rosanka et al., ACP, 2020 should be mentioned already here and not only in l. 323, as their findings appear to result also from actual weather patterns (related to the same project?)

We added a paragraph explaining the variety of REACT4C related studies and their particular focus and how the studies relate. We also mentioned the companion paper of Rosanka in that paragraph (l. 100).

l. 107: The entire Section 2 forms a dense amount of text which is difficult to digest as one block. I suggest structuring it more clearly by breaking into subsections. I would suggest at least a section describing the model components, and a further section describing the set up used for the experiments and how the model outputs are obtained (combination of Eulerian and Lagrangian approach?). A dedicated description of how the experiments were conducted is missing, instead it seems to be scattered over the remaining parts of the manuscript. Please draw this together in a clearly reproducible experiment description.

Thank you for pointing this out. We have added extensive and more detailed description of the methodology and broke up the section in several subsections as suggested. In addition, we implemented a subsection describing the experimental set up (2.5).

l. 131: Why is 00 UTC not part of the emission times?

Night flying restrictions are common at airports in Europe, many airports have restrictions and curfews during the night. As most departures start at 6 or 7 in the morning, as would the optimization of flights, we saw no need for CCFs in the period with restricted or banned flights. However, in a follow-up project, we calculated CCFs for 5 times within 32 hours covering also 0:00 UTC (twice) to be able to theoretically optimize flights from EU to USA and return within one set of CCFs.

On what criteria have the authors chosen the magnitude of the NOx and H2O emission perturbations, are they in any way related to an emissions inventory? Particularly the H2O value looks oddly precise. Also, are these perturbations in addition to the existing aircraft emissions at the grid point location?

The NOx and H2O emission perturbations within one time-region have deliberately been amplified compared to the background aviation emissions, so that the pulse emission for one timestep remains detectable. We added information on the background emissions assumed (l. 259f).

Table 1: Thinking about the background conditions at these grid point locations: How do they overlap with the present flight routing/aircraft emissions distribution? Are they all located inside the main North Atlantic flight corridor?

The time-regions cover the region where most transatlantic flights occur (maximum between 40-55°N), plus additional time-regions further north and south (30°N and 80°N), so that potential latitudinal rerouting of aircraft trajectories can be investigated. The four pressure levels cover the main flight levels and levels below.

l. 207: In the interest of understanding, clarity and reproducibility, it is not clear how the perturbations were applied. Obviously, this study consisted of a substantial number of model experiments (as indicated by the total CPU hours), however the 280 experiments remain still an order of magnitude below the 4032 perturbation events. How were they grouped together? How many perturbations were applied during one experiment? How many experiments represent one weather situation? Further, after a detailed listing of the model's abilities and sub-components (with citation) in Section 2 of the manuscript, there is a lack of technical detail how the experiments were conducted. Were they all initialised from a common set of initial conditions, perhaps one for each weather situation? How long was the model spun up for prior to the 90 days of the output that was taken into consideration?

We agree, that a number of details were missing to be able to follow the realization of the experiments and in fact a set of perturbation events were included in every experiment. We have added detailed information on all the points mentioned above (Section 2.5).

Figures 2 and 3: It is not clear how these findings relate to the experiments: Is this the additive result of all the perturbations combined? How do you obtain from perturbations in individual grid cells, which are propagated through trajectories, a regional map of potential contrail coverage in %? This should become clear through an improved experiment description in Section 2. Similarly, this will also help interpreting Figs 7, 10 and 11. While the method of compiling the information needs better explaining, I would like to point out that I find the information conveyed by these maps of great value and I welcome this method of visualising the results in general (and I am glad that figure labelling has a reasonable, legible size!)

Figure 2 (now 3) shows the background atmospheric condition to be able to form contrails at the time of emission release for every weather situation. It was meant to facilitate better understanding of Figure 7, i.e. CCFs in Figure 7 can only be nonzero, where potential contrail coverage is nonzero in Figure 3. Figure 3 was meant to illustrate this relationship. Furthermore, it was meant to illustrate the characteristics and spatial distribution of potential contrail coverage for the different weather situations. We have added some sentences to the respective paragraphs to make this clearer. Furthermore we changed the Figure showing a temporal mean potential contrail coverage (6, 12, 18 UTC) to only 12 UTC, so that the relation to the CCF is more straightforward.

The Figures 3 to 6 enable comparison of our findings to other studies and these are helpful to evaluate the model capabilities, as to our knowledge no study calculated contrails and contrail-cirrus and their properties on air parcel trajectories before.

Section 4.2: My concern here stems again mainly from a lack of understanding how to interpret the figures. While contrails are formed almost instantly in the wake of the aircraft, impacts from NOx emissions have a longer lifetime and the map representation in Figs 7 and 10 only makes sense if it indicates the location of the emissions release (even though the climate change impact might be felt elsewhere). Can the authors confirm this?

Your assumption is correct, these figures designate the impact mapped back at the specific emission location and refer to the global effect an emission would have if released at this time-region (even if the effect reveals elsewhere). We added a few sentences to clarify this, to ensure the reader understands this interrelationship and is able to interpret the Figures correctly (l. 297ff).

l. 379: It would be interesting to know the author's view on why the jet stream region would yield generally more positive values, seeing that otherwise subtropical high pressure belts exhibit positive values.

This is indeed a very interesting aspect. When investigating the high $O_3$-CCF values of W1 and W4 in the core of the jet stream, we find that the air parcels emitted in this region are mainly transported towards lower latitudes and lower altitudes. In the companion paper by Rosanka et al. 2020, we find that air parcels that are transported to lower latitudes and altitudes are transported into regions with a high chemical efficiency, which results in a higher potential climate impact. In the revised manuscript, we added a few sentences explaining this relation (l. 431ff).

l. 382: Here the authors compare their findings with climatological studies that were partly carried out with even lower grid resolution (e.g. T21). Previously the authors have shown the importance of the pathway of NOx emissions which would be better resolved with higher model grids. To what extent do the authors believe a T42 representation of short-lived weather situations offers a fair comparison to climatological studies? While I can understand that it can instil confidence to have findings agree with previous studies, I would like to be reassured that the comparison is on fair terms? How important do the authors consider grid resolution in their study? What is the minimum resolution required to resolve the weather situations sufficiently to represent the emissions transport on non-climatological time scales?

In the past, many studies suffered from limitations in computational resources. We agree, that a comparison to these studies might not be straightforward and we would refrain from evaluating our study by using coarse resolution studies, however to check whether the big picture makes sense, makes it worthwhile mentioning them. We added an additional newer study by Lund et al., 2017 which also employed a T42 horizontal resolution. They find largest effects of aviation NOx for emissions in the tropics and subtropics and smaller effects for emissions at midlatitudes. However, they point out, that

the sign and magnitude of total NOx climate effects depend strongly on the chosen metric and time horizon but also on inter-model differences in the change in methane lifetime per unit ozone change as stated by Köhler et al. (2013) and Myhre et al.,(2011). Within our study we could not gain any information on the minimum resolution required to resolve weather situations sufficiently. However, we can confirm, that in our study the transport pathways are well represented and weather patterns compare very well with the classification using ERA-Interim data (Irvine et al., 2013). We adapted the text to discuss these aspects and refer to the newer study by Lund et al. (2017). (l. 496ff)

l. 413: explicitly (typo)

Corrected.

Conclusions: The first paragraph is fairly high level, only telling me that jet stream location, tropopause height, high pressure systems and polar night are all having an influence, but this is not further elaborated. It would be nice if the main findings or messages could be brought more clearly across. Is there a pattern that the authors recognise? If the results are too diverse to summarise then perhaps it should be stated here.

We agree, that the take away message could be formulated more specifically. We added some sentences to the conclusions to hopefully improve this section (l. 588ff)

l. 469 or l. 479: As previously mentioned, the conclusions here need to acknowledge the limits in grid resolution and what impact the authors believe this might have on their findings.

We added this aspect to the conclusion section (l. 593ff). Within our study we could not gain any information on the minimum resolution required to resolve weather situations sufficiently. However, we can confirm, that in our study the transport pathways are well represented and weather patterns agree well with the classification using ERA-Interim data (Irvine et al., 2013). See also the discussion on resolution topics above.

---

## Author Comment (AC2) · 5 Mar 2021

The paper address non-CO2 climate impact of aviation, quantifies and presents climatology of these effects for the North Atlantic and discuss possibilities for mitigation of these impacts through alternative routing. This is a very complex scientific question relevant for ACP, also very topical as the non-CO2 climate impacts, if unresolved, makes it difficult for the aviation sector to achieve the Paris agreement targets. The paper present results of model simulations where the local aircraft emissions are followed in a global ECHAM5/MESSy Atmospheric Chemistry model system on Lagrangian trajectories. There are several novel modules developed within the model system associated with these calculations which have been published and are cited as accompanying papers, the most important Grewe et al. (2014) and Rosanka et al. (2020). While Grewe et al. (2014) presents results for 1 case (weather situation), this paper presents results for a number of situations under winter and summer season which allows for more general conclusions based on thousands of simulated trajectories and their complex analysis. These papers represent an impressive piece of work which gives a new insight in climatology of non-CO2 climate effects of aviation and I would like to congratulate the authors to this achievement.

We thank the anonymous referee for seeing the value of our work and estimating our work as relevant for ACP. We gratefully appreciate his/her time reading our manuscript thoroughly and pointing towards paragraphs which were not clear enough or needed improvement. We amended the manuscript accordingly. Please find our responses (red) to the review points and recommendations (black) below.

Before recommending the paper for publication, I would however like to draw their attention to the following issues:

The methods and assumptions in the paper are extremely complex and described in many cases rather briefly with references to other papers. This makes it rather difficult to follow and assess the methodology. I would recommend including more comprehensive and systematic description of the methodology which would give clear idea about how the crucial processes are treated in the model simulations performed for this paper.

Thank you for pointing this out. We agree, that the description for such a complex set up was too brief to follow and assess the methodology. We added extensive parts and more details to the methodology description, also proposing a substructure of Section 2 in order to increase the comprehensibility of this essential part of the manuscript.

Even when going to the references I could not follow how the chemistry of the 'multitudes of background trajectories' which the local emission trajectory is mixed with is calculated, recommend explanation of this part in particular.

We agree, that this aspect was not covered in the methodology description. We added this and we hope it is now clearer (l. 156ff).

After reading the papers describing the tagging mechanism for quantification of impact of studied emissions, I would like to ask if the non-linear plume effects on ozone formation from NOx emissions, or rather on NOx removal, are considered in some way and, as this has been subject of scientific discussion under quite a long time, what impact these effects have or could have in case they were not considered).

This aspect was not considered in the present study. Plume effects on ozone formation have been studied by several groups, however the results vary considerably, ranging from -33% to +5% compared with the

grid-scale approach (e.g. Cameron et al., GRL 2013). This effect might be largest in the North Atlantic Flight corridor (Cariolle et al., 2009, JGR). However, as the results of the variety of studies on these effects are not yet conclusive, we did not consider non-linear plume effects on ozone formation. Nevertheless, Grewe et al 2014b performed a sensitivity study on the cost-benefit analysis and related air traffic responses with respect to potential errors in the climate-change functions. For example, they changed the weighting between the climate impact from NOx and contrail-cirrus. In this case, the air traffic system still behaves similarly during the optimization. These sensitivity studies indicate a stable response in the shape of the cost benefit analyses and the way air traffic is routed for climate impact. We consider this point to be covered through the uncertainty/sensitivity studies performed by Grewe 2014b by means of these data. We added more detailed information on these sensitivity studies in the discussion (l. 533ff).

Specific comments: Fig. 2 – Potential contrail coverage for the representative weather situations – what is the figure showing – mean over certain time period of each weather situation? Supplement – Caption or some explanation to the figures is missing

This information was indeed missing in the figure caption. The original Figure was showing mean potential contrail coverage for the three emission time steps 6, 12 and 18 UTC, however, we decided that it is more helpful to show 12 UTC only to better estimate the relation of potential contrail coverage at the time step of emission release to the CCFs shown in Figure 7.

---

## Author Response (AR2)

April 30, 2021

Dear Editor Prof. Jayanarayanan Kuttippurath,

We are pleased to hear, that our manuscript is accepted for publication and we gratefully acknowledge your support during the review process.

During the preparation of production files we found the following items, which we ask for permission to amend before the official publication and before we upload the production files. These changes are not critical to the outcome, and the conclusions of the manuscript remain unchainged. Please find the items we would like to change, listed below (the numbers of pages and lines refer to the attached file: *diff_to_acp-2020-529-froemming_revised.pdf*):

1. Include one reference, which was published in the meantime (Page 3, Line 69 and Line 83).
2. Shorten one very long sentence (Page 8, Line 227).
3. Add and sharpen a few sentences in the model description (Page 9, Line 237-253).
4. Update of the constant conversion factor for contrail CCFS in Figure 7 (Page 15, Figure 7) and in Figures S1-S8 in the supplementary material.
5. Update of the Units in Figures 8 and 11 (the Unit is $[K/kg(NO_2)]$ instead of $[K/kg(N)]$) (Page 16, Figure 8 and Page 20, Figure 11) and in Figures S9-S16 in the supplementary material.
6. Update of Figure 16 due to updated constant conversion factor of contrail CCFs (see Point 4 above). We propose to show weather situation S2 instead of W4 as it better illustrates the dipole structure with warming and cooling regions. The text remains with only slight modifications. (Page 25, Line 565 and Lines 559f; Page 26, Figure 16).
7. Acknowledge helpful discussions with a colleague (Page 29, Line 672).

Please accept our apologies for these late changes and any inconveniences these may cause. All the amendments we suggest are not critical to the main points, results and consequences of the manuscript therefore we hope for your authorization.

Thank you for your time and consideration,

Dr. Christine Frömming (on behalf of all co-authors)